# A Constrained Optimization Perspective of Unrolled Transformers

## Abstract

We introduce a constrained optimization framework for training transformers that behave like optimization descent algorithms. Specifically, we enforce layerwise descent constraints on the objective function and replace standard empirical risk minimization (ERM) with a primal-dual training scheme. This approach yields models whose intermediate representations decrease the loss monotonically in expectation across layers. We apply our method to both unrolled transformer architectures and conventional pretrained transformers on tasks of video denoising and text classification. Across these settings, we observe constrained transformers achieve stronger robustness to perturbations and maintain higher out-of-distribution generalization, while preserving in-distribution performance.

## 1 Introduction

Unrolling arises from the observation that iterations of descent algorithms of some optimization problems perform operations that are analogous to those of individual layers of a neural network (Gregor and LeCun, 2010; Monga et al., 2021). From this observation, an extensive literature has emerged in which neural networks are trained to solve optimization problems, with corresponding descent algorithms used as guidance for architecture design (Yang et al., 2022; De Weerdt et al., 2023; Yang et al., 2021; Xie et al., 2023; Hershey et al., 2014; Frecon et al., 2022). E.g., descent algorithms for sparse reconstruction involve a linear map and a nonnegative projection motivating the use of a neural network made up of linear maps and ReLU nonlinearities to learn solutions of sparse reconstruction problems (Gregor and LeCun, 2010).

In the case of transformers (Vaswani et al., 2017), unrolling has gained traction as a tool to interpret attention mechanisms (Yang et al., 2022; Yu et al., 2023; Ramsauer et al., 2020; De Weerdt et al., 2024; Von Oswald et al., 2023). These works present different energy functions and provide theoretical results showing that the update rules, which closely resemble a transformer's forward pass, exhibit descent properties. Beyond this theoretical value, (De Weerdt et al., 2023) and (Yu et al., 2023) train unrolled transformers that are interpretable and parameter-efficient. When training these models, however, the behavior of these networks is non-monotonic along the iterates, which is inconsistent with the behavior expected of an optimizer. There are two reasons for this. Firstly, there is no guarantee that the learned parameters will satisfy the conditions under which the models minimize the energy. Secondly, training an unrolled architecture inherently induces a bilevel problem, which the unrolled architecture is not designed to take into account.

In this paper, we draw from the unrolling literature to argue that it may be advantageous to train transformers that *behave* like descent algorithms. We do so by imposing descent constraints on the output of each layer. Specifically, the first contribution of this paper is that:

**[C1]** We formulate a constrained learning problem in which the output of each layer of a transformer is required to reduce the expected loss by a given factor relative to the cost of the output of the previous layer (Section 2).

It is important to point out that our use of the term unrolling is not identical to the more common use of unrolling to refer to a learned parameterization that solves an optimization problem. We use unrolling here to refer to an arbitrary learning problem in which we explore the merit of forcing the layers of the transformer to behave like steps of an optimization descent algorithm.

We discuss training algorithms for constrained transformers in Section 3. These algorithms train transformers in the dual domain where we leverage small duality gap results drawn from the constrained learning literature (Chamon et al., 2023; Chamon and Ribeiro, 2020). This is our second contribution:

**[C2]** We develop a dual training algorithm for constrained unrolled transformers. We show that incorporating descent constraints ensures asymptotic convergence–in the number of layers–to a near-optimal value of the

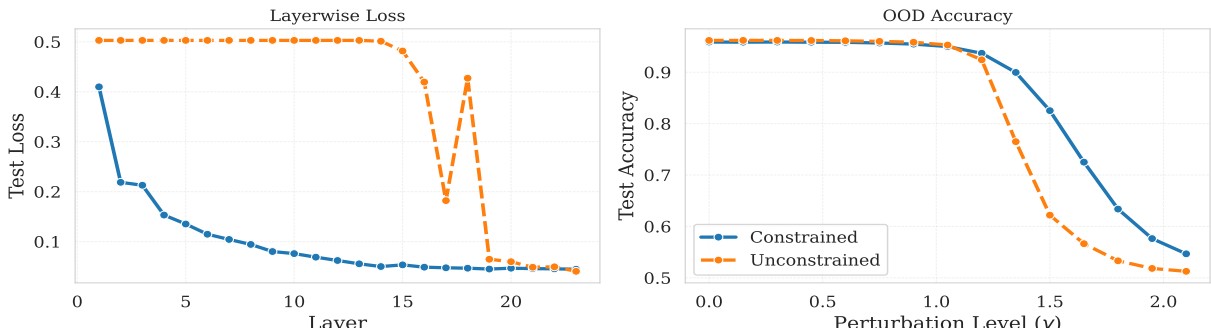

Figure 1: **Layerwise descent improves OOD robustness.** Left: Test loss at each layer (↓ lower is better). Constrained RoBERTa exhibits monotonic descent, unlike the unconstrained baseline. Right: Out-of-distribution accuracy under increasing embedding perturbation levels $\gamma$ (↑ higher is better). As $\gamma$ grows, the constrained model degrades more gracefully and retains higher accuracy. Setting: RoBERTa ($L = 24$) trained on IMDb, training $\gamma = 0.2$.

statistical loss. We also provide theoretical *guarantees* that constrained transformers maintain descent behavior and exhibit out-of-distribution (OOD) generalizability under distribution shifts (Section 3).

We expect the incorporation of descent constraints to yield trained transformers that respond better to perturbations of the input data. This is because it is a hallmark of descent algorithms that they *do* respond better to perturbations and existing results have shown that incorporating descent constraints in neural networks does result in learned solutions that are less sensitive to perturbations (Hadou et al., 2024b). In this paper, we find that transformers trained with descent constraints share this property. Our next two contributions are to demonstrate the value of adding constraints in two application domains where data perturbations arise naturally:

**[C3]** We consider video denoising, where the goal is to learn a transformer for vision (Yang et al., 2022; De Weerdt et al., 2023; Dosovitskiy et al., 2021) that recovers a video sequence from noisy observations. We show that training with descending constraints results in models that are more robust to OOD levels of noise, as measured by the root mean squared error (RMSE) of the reconstructed videos (Section 4).

**[C4]** We investigate text classification problems with perturbed embeddings, in which users query a language model with embeddings perturbed with varying levels of Gaussian noise (Fukuchi et al., 2017). We observe that the reduction of classification accuracy as the degree of perturbation increases is smaller in transformers trained with descent constraints (Section 5).

An instance of this robustness to perturbations is illustrated in Figure 1. A text classifier (RoBERTa) trained with empirical risk minimization (ERM) exhibits a non-monotonic loss pattern along its layers, maintaining a high loss until the last few layers, with a spike at the 19th layer. The constrained version of this model, trained to enforce monotonic descent constraints, exhibits a smoother decreasing pattern. When measuring the accuracy under OOD perturbations, we observe that the constrained model's accuracy decays more gracefully than its unconstrained counterpart.

## 1.1 RELATED WORK

**Algorithmic unrolling and the learning-to-optimize framework.** There exists a vast literature in algorithmic unrolling, in which a neural network learns to approximate the solution of an optimization algorithms (Monga et al., 2021). Since the original LISTA (Gregor and LeCun, 2010), subsequent works have obtained remarkable improvements in speed and performance (Liu et al., 2019a; Chen et al., 2018; Aberdam et al., 2020) and unrolling has been applied to learn approximations of a variety of optimization algorithms (Hershey et al., 2014; Sprechmann et al., 2015; Wang et al., 2015).

**Other unrolled neural networks.** Beyond traditional unrolling, a new line of research has emerged that uses unrolling as a theoretical tool to interpret a variety of neural network architectures and layers, such as graph neural networks (GNNs) (Yang et al., 2021; Hadou et al., 2024a; Hadou and Ribeiro, 2025), recurrent neural networks (RNNs) (Luong et al., 2021), ReLU nonlinearities (Xie et al., 2023), and other feedforward architectures (Frecon et al., 2022).

**Unrolled transformers.** The first work to show an unrolling of an attention layer is (Ramsauer et al., 2020). The first full unrolling of a transformer layer with attention and a nonlinearity is attributed to (Yang et al., 2022). This is

extended by (De Weerdt et al., 2023) for video reconstruction with a LISTA nonlinearity instead of ReLU. The work in (Yu et al., 2023) is a different unrolling that interprets the transformer as a process of denoising and compressing tokens.

**Transformers as optimizers.** Simultaneously, the community has explored other perspectives of transformers as optimizers. One such view is that transformers are in-context learners (Dong et al., 2024; Oswald et al., 2023; Li et al., 2023; Olsson et al., 2022; Ahn et al., 2023; Dai et al., 2022), which may explain large language model's abilities to generalize to tasks not seen during training via examples provided during inference (Brown et al., 2020).

**Constrained learning and constrained unrolling.** Constrained learning theory provides a framework for training neural networks subject to constraints (Chamon et al., 2023; Hounie et al., 2023) and has been a useful tool in various domains (Moro and Chamon, 2024). In the context of unrolling, (Hadou et al., 2024b;a) have proposed training unrolled networks with descent constraints and demonstrated that constrained models exhibit more robustness to perturbations and better out-of-distribution generalization.

**Differential privacy.** Additive Gaussian perturbations are common in Differential Privacy (DP) for neural networks (Dwork and Roth, 2014; Dwork et al., 2006; Yu et al., 2022). One method to train private models is to perturb the gradients during training (Abadi et al., 2016), which gives $(\epsilon, \delta)$-DP. Another approach, closer to our experimental case, is to perturb inputs directly, known as *local*-DP. This mechanism ensures end-to-end user privacy but leads to even worse $O(\sqrt{n}\epsilon, \delta)$ privacy (Fukuchi et al., 2017) and thus requires more noise for the same privacy level. Other DP works also study perturbing token embeddings (Yu et al., 2022; Feyisetan et al., 2020; Feyisetan and Kasiviswanathan, 2021; Bollegala et al., 2023).

## 2 CONSTRAINED UNROLLED TRANSFORMERS

A transformer is a layered architecture that processes a sequence of $T$ vectors $\mathbf{x}_t \in \mathbb{R}^N$ grouped in the matrix $\mathbf{X} = [\mathbf{x}_1, \ldots, \mathbf{x}_T] \in \mathbb{R}^{N \times T}$ to represent the entire vector sequence. The input to each transformer layer is a matrix $\mathbf{X}_{l-1} = [\mathbf{x}_{l-1,1}, \ldots, \mathbf{x}_{l-1,T}] \in \mathbb{R}^{N \times T}$ and the output is another matrix $\mathbf{Y} = [\mathbf{y}_1, \ldots, \mathbf{y}_T] \in \mathbb{R}^{N \times T}$, both of which are also sequences with the same dimensions as $\mathbf{X}$. The first component of a transformer layer is an attention operation whose output is a matrix $\mathbf{Z}$ given by

$$\mathbf{Z}_l = \mathbf{V}_l \mathbf{X}_l \times \text{sm}\left[ \left(\mathbf{Q}_l \mathbf{X}_{l-1}\right)^T \left(\mathbf{K}_l \mathbf{X}_{l-1}\right) \right] = \mathbf{V}_l \mathbf{X}_{l-1} \times \text{sm}(\mathbf{A}_l). \tag{1}$$

In (1), the matrix $\mathbf{Z}_l = [\mathbf{z}_{l1}, \ldots, \mathbf{z}_{lT}] \in \mathbb{R}^{D \times T}$ represents a sequence of vectors $\mathbf{z}_{lt} \in \mathbb{R}^D$ with dimension $D$ typically much smaller than $N$. The matrices $\mathbf{Q}_l, \mathbf{K}_l, \mathbf{V}_l \in \mathbb{R}^{D \times N}$ are called query, key, and value matrices and are trainable parameters. The matrix $\mathbf{A}_l := (\mathbf{Q}_l \mathbf{X}_{l-1})^T (\mathbf{K}_l \mathbf{X}_{l-1})$ is a linear attention matrix and the operation $\text{sm}(\mathbf{A}_l)$ acts separately on rows of $\mathbf{A}$ so that if $\mathbf{B} = \text{sm}(\mathbf{A})$ we have $b_{lut} = \exp(a_{lut}) / \sum_{t=1}^T \exp(a_{lut})$.

The second operation in a transformer involves matrices $\mathbf{W}_l \in \mathbb{R}^{N \times D}$ and $\mathbf{U}_l \in \mathbb{R}^{N \times N}$ as trainable parameters and a pointwise nonlinear function $\sigma$ and entails the processing of the time series $\mathbf{Z}_l$ with a linear perceptron that also includes a residual connection of the layer's input vector $\mathbf{X}_{l-1}$,

$$\Phi_l(\mathbf{X}; \mathbf{T}) = \mathbf{Y}_l = \sigma\left[\mathbf{W}_l \mathbf{Z}_l + \mathbf{U}_l \mathbf{X}_{l-1}\right]. \tag{2}$$

The sequence $\mathbf{Y}_l = \Phi_l(\mathbf{X}; \mathbf{T})$ is the output of layer $l$. A transformer is defined by $L$ recursive applications of (1)–(2) by making the output of layer $l$ the input to layer $l+1$, i.e., $\mathbf{X}_{l+1} = \mathbf{Y}_l$. The input to layer 1 is the given sequence $\mathbf{X}_0 = \mathbf{X}$ and the output of the transformer is the output of layer $L$, $\mathbf{Y}_L = \Phi_L(\mathbf{X}; \mathbf{T})$. We write the output of layer $l$ as a function $\Phi_l(\mathbf{X}; \mathbf{T})$ of the input sequence $\mathbf{X}$ and the trainable tensor $\mathbf{T}$ which groups the matrices $\mathbf{Q}_l, \mathbf{K}_l$ and $\mathbf{V}_l$ of (1) as well as the matrices $\mathbf{W}_l$ and $\mathbf{U}_l$ of (2) for all layers $l$.

Consider now a loss function $f(\mathbf{X}, \Phi(\mathbf{X}; \mathbf{T}))$ dependent on the input and output values of the transformer. It is customary to seek parameters $\mathbf{T}_U^*$ that minimize the average loss,

$$\mathbf{T}_U^* = \underset{\mathbf{T}}{\text{argmin}} \, \mathbb{E}\left[ f\left( \mathbf{X}, \Phi(\mathbf{X}; \mathbf{T}) \right) \right]. \tag{3}$$

In prior contributions, it has been observed that transformers can be interpreted as iterative descent algorithms that solve optimization problems (Yang et al., 2022; De Weerdt et al., 2023; Yu et al., 2023; Ramsauer et al., 2020). In this paper, we draw inspiration from this idea and argue that it may be advantageous to train transformers that behave like iterative descent algorithms. Formally, we consider a stepsize schedule $0 < \alpha_l < 1$ and propose to train transformers that solve

the constrained learning problem,

$$\mathbf{T}^* = \underset{\mathbf{T}}{\operatorname{argmin}} \quad \mathbb{E}\Big[ f\big( \mathbf{X}, \Phi(\mathbf{X}; \mathbf{T}) \big) \Big],$$

$$\text{subject to } \mathbb{E}\Big[ f\big( \mathbf{X}, \Phi_l(\mathbf{X}; \mathbf{T}) \big) \Big] \leq (1 - \alpha_l) \, \mathbb{E}\Big[ f\big( \mathbf{X}, \Phi_{l-1}(\mathbf{X}; \mathbf{T}) \big) \Big], \, \forall l. \tag{4}$$

The purpose of the constraints in (4) is to force the optimal transformer $\mathbf{T}^*$ to have layers that reduce the statistical loss $f$ progressively. Since this is a property of descent algorithms, we say that $\mathbf{T}^*$ is an unrolled transformer. We point out, however, that this is not an exact analogy to the standard use of the term unrolling which involves the use of neural networks or transformers to *solve* optimization problems (Monga et al., 2021; De Weerdt et al., 2023)—rather than encouraging a transformer to *descend* like optimization algorithms do. It is also worth noting that analogous formulations to (4) can be derived for other deep neural network architectures, for which the theoretical results of this work would similarly apply.

## 3 TRAINING OF CONSTRAINED UNROLLED TRANSFORMERS

Problem (4) involves finding the transformer parameters $\mathbf{T}^*$ that minimize the loss function $f$ subject to the descent constraints. This formulation is a nonconvex constrained problem, which is usually difficult to solve directly. Rather, we resort to the dual problem, constructed through the Lagrangian function,

$$\mathcal{L}(\mathbf{T}, \boldsymbol{\lambda}) = \mathbb{E}\Big[ f(\mathbf{X}, \Phi(\mathbf{X}; \mathbf{T})) \Big] + \sum_{l=1}^{L} \lambda_l \mathbb{E}\Big[ f(\mathbf{X}, \Phi_l(\mathbf{X}; \mathbf{T})) - (1 - \alpha_l) f\big( \mathbf{X}, \Phi_{l-1}(\mathbf{X}; \mathbf{T}) \big) \Big], \tag{5}$$

where the vector $\boldsymbol{\lambda} \in \mathbb{R}_+^L$ collects the Lagrangian multipliers. The dual problem is then defined as

$$\widehat{D}^* = \max_{\boldsymbol{\lambda}} \min_{\mathbf{T}} \widehat{\mathcal{L}}(\mathbf{T}, \boldsymbol{\lambda}), \tag{6}$$

where $\widehat{\mathcal{L}}$ is the empirical Lagrangian function, evaluated over $M$ realizations of $\mathbf{X}$. The max-min problem in (6) can be viewed as a sequence of regularized ERM problems, solved sequentially, differing only in the choice of Lagrangian multipliers. There is empirical evidence that a *high-quality* local minimum for such unconstrained problems can be attained using stochastic gradient descent (Zhang et al., 2016; Arpit et al., 2017). The Lagrangian multipliers, which act as regularization parameters in this view, are updated using projected gradient ascent to maximize the dual function, since the latter is concave. Solving the dual problem then entails alternating between minimization with respect to $\mathbf{T}$ and maximization over $\boldsymbol{\lambda}$ (Chamon and Ribeiro, 2020; Fioretto et al., 2021), leading to the primal-dual procedure described in Algorithm 1.

Although classical duality theory (Boyd and Vandenberghe, 2004) indicates that nonconvex constrained programs may exhibit non-zero duality gaps, recent results show that, in training deep neural networks, this duality gap is typically small (Chamon et al., 2023). We include these results in the following theorem to keep our discussions self-contained.

**Theorem 1** (Constrained Learning Theorem (Chamon et al., 2023))**.** *Let* $(\mathbf{T}^*, \boldsymbol{\lambda}^*)$ *be a stationary point of* (6) *and* $P^*$ *denote the optimal value of the statistical loss function in* (4)*. Under Assumptions 1 - 5 (see Appendix A.1), it holds, for some constant* $\rho$*, that*

$$|P^* - \widehat{D}^*| \leq C\nu + \rho\,\zeta(M, \delta), \quad and$$

$$\mathbb{E}\Big[ f\big( \mathbf{X}, \Phi_l(\mathbf{X}; \mathbf{T}^*) \big) \Big] - (1 - \alpha_l) \, \mathbb{E}\Big[ f\big( \mathbf{X}, \Phi_{l-1}(\mathbf{X}; \mathbf{T}^*) \big) \Big] \leq \zeta(M, \delta), \, \forall l,$$

*with probability* $1 - \delta$ *each, and with* $\rho = \max\{\|\boldsymbol{\lambda}^*\|, \|\bar{\boldsymbol{\lambda}}^*\|\}$*, where* $\bar{\boldsymbol{\lambda}}^*$ *is the optimal multiplier of the statistical dual problem. Moreover,* $\nu$ *and* $\zeta(M, \delta)$ *are the expressivity parameter and the sample complexity, respectively, and* $C$ *is a Lipschitz constant.*

The theorem affirms that (6) yields near-optimal near-feasible solutions to (4) and can replace it. This result stems from the fact that the functional version of (4) exhibits zero duality gap. When we optimize over an *expressive* parameterized class, we incur an optimality loss that amounts to the expressivity parameter $\nu$. However, there is theoretical evidence that $\nu$ can be made arbitrary small in deep neural networks (Ryu et al., 2019; Graikos et al., 2022) and also in transformers (Yun et al., 2020). The second source of error is the empirical approximation of the Lagrangian function, quantified by the sample complexity $\zeta(M, \delta)$, and can be reduced by increasing the sample size $M$.

Theorem 1, however, makes it challenging to conclude converges guarantees, since it provides only high-probability near-feasibility guarantees. That is, even though (4) requires each layer to enforce descent in $f$, the near-feasible

---

**Algorithm 1** Primal-dual Training Algorithm for Constrained Unrolled Transformers

---

1: *Inputs*: number of epochs, batch size $M$, step sizes $\eta_1, \eta_2 > 0$
2: *Initialize*: $\mathbf{T}, \boldsymbol{\lambda} = \{\lambda_\ell\}_{\ell=1}^{L}$
3: **for** each epoch **do**
4:     **for** each batch of samples $\mathbf{X} = \{\mathbf{X}^{(m)}\}_{m=1}^{M}$ **do**
5:         Execute the forward pass $\Phi(\mathbf{X}^{(m)}; \mathbf{T})$ according to (1) and (2) for all $m$
6:         Compute the Lagrangian $\widehat{\mathcal{L}}(\mathbf{T}, \boldsymbol{\lambda})$
7:         Primal update: $\mathbf{T} = \mathbf{T} - \eta_1 \nabla_{\mathbf{T}} \widehat{\mathcal{L}}(\mathbf{T}, \boldsymbol{\lambda})$
8:         Dual update: $\boldsymbol{\lambda} = \left[ \boldsymbol{\lambda} + \eta_2 \nabla_{\boldsymbol{\lambda}} \widehat{\mathcal{L}}(\mathbf{T}, \boldsymbol{\lambda}) \right]_+$
9:     **end for**
10: **end for**
11: **Return** $\mathbf{T}^* = \mathbf{T}$.

---

solution satisfies each constraint with probability $1 - \delta$. The probability that all $L$ constraints are satisfied is then $(1 - \delta)^L$. For large $L$, this probability could be small, implying that this theorem alone is insufficient to establish the required layerwise descent and convergence guarantees.

In Theorem 2, we show that the constrained unrolled transformer, obtained by (6), converges asymptotically–in the number of layers–to the optimal value of the statistical loss.

**Theorem 2** (Convergence Guarantees). *Given a constrained unrolled transformer $\mathbf{T}^*$, which satisfies Theorem 1, and a functional minimizer $\mathbf{Y}^*$ of the statistical loss. Let $\alpha_l = \alpha$, for all $l$. Then, under Assumption 6 (see Appendix A.2), it holds that*

$$\lim_{l \to \infty} \min_{k \leq l} \mathbb{E}\left[ f\left( \mathbf{X}, \Phi_k(\mathbf{X}; \mathbf{T}^*) \right) - f\left( \mathbf{X}, \mathbf{Y}^* \right) \right] \leq \frac{1}{\alpha} \left( \zeta(M, \delta) + \frac{C\delta\nu}{1 - \delta} \right) \quad a.s. \tag{7}$$

Theorem 2 states that the constrained unrolled transformer is guaranteed to attain the optimal performance up to an error that is controlled by the step size $\alpha$, the sample size $M$ and the expressivity of the model class $\nu$. The proof proceeds by showing that, despite the aforementioned probabilistic constraint violations, the sequence of layer losses forms a supermartingale and converges infinitely often to a sub-optimal region, characterized by the bound. The full proof of Theorem 2 is relegated to Appendix A.2. The sample complexity controls the failure probability $\delta$, and for sufficiently large $M$, we can keep $\delta$ arbitrary small and eliminate the second term of the bound. Moreover, large $\alpha$, which corresponds to aggressive reductions, shrinks the size of the sub-optimal region and provides tighter guarantees. Although this result holds asymptotically, our numerical results demonstrate that we can achieve the same performance of a *good* local minimizer of (3) in a finite number of layers.

Imposing layerwise constraints endows transformers with monotone-descent inductive biases. Such descent properties provide classical optimization methods with stability under perturbations and generalization across problem instances. By aligning the transformer's layer-to-layer dynamics with these properties, the transformer maintains comparable performance under distribution shifts. The following corollary and theorem formalize this effect and establish OOD generalization guarantees for transformers trained with descent constraints.

**Corollary 3.** *Let $\mathbf{T}^*$ be a constrained unrolled transformer trained on a data distribution $D_x$. Then, for any shifted distribution $D_{x'}$ that satisfies Assumption 7 (see Appendix A.3), it holds with probability $1 - \delta$, for all $l$:*

$$\mathbb{E}_{D_x'}\left[ f\left( \mathbf{X}, \Phi_l(\mathbf{X}; \mathbf{T}^*) \right) \right] - (1 - \alpha_l) \mathbb{E}_{D_{x'}}\left[ f\left( \mathbf{X}, \Phi_{l-1}(\mathbf{X}; \mathbf{T}^*) \right) \right] \leq \zeta(M, \delta) + C\tau, \tag{8}$$

*where $\tau = d(D_x, D_{x'}) + d(D_{x'}, D_x)$, and $d(\cdot, \cdot)$ is a bounded asymmetric distance metric.*

**Theorem 4** (Out-of-Distribution Guarantees). *Let $\widehat{\mathbf{Y}}^*$ be a functional minimizer of the statistical loss evaluated on $D_{x'}$. Then, the constrained unrolled transformer trained on $D_x$ satisfies*

$$\lim_{l \to \infty} \min_{k \leq l} \mathbb{E}_{D_{x'}}\left[ f\left( \mathbf{X}, \Phi_k(\mathbf{X}; \mathbf{T}^*) \right) - f\left( \mathbf{X}, \widehat{\mathbf{Y}}^* \right) \right] \leq \frac{1}{\alpha} \left( \zeta(M, \delta) + C\tau + \frac{C\delta\nu}{1 - \delta} \right). \tag{9}$$

The proofs are in Appendices A.3 and A.4. Corollary 3 states that a constrained unrolled transformer trained on one distribution also satisfies the descent constraints under a shifted distribution $D_{x'}$ up to an additional error proportional to the distance between the two distributions. Then, it follows that the constrained unrolled transformer converges to the optimal of the statistical loss evaluated on $D_{x'}$ up to the same additional error bound, as formalized in Theorem 4.

The assumptions under which our theoretical results hold are readily achievable in practice (see Appendix A.1). However, the feasibility assumption is not widely guaranteed, as it requires the constrained problem (4) to be strictly feasible. To enforce this condition, we consider a resilient constrained learning relaxation, as proposed in (Hounie et al., 2023), wherein we introduce an additive slack variable $\mathbf{u} \in \mathbb{R}_+^L$ into the constraints and augment the loss function $f$ with a quadratic penalty term $\frac{\beta}{2}\|\mathbf{u}\|_2^2$. This transforms the saddle point problem into the regularized formulation:

$$\max_{\boldsymbol{\lambda}} \min_{\mathbf{T}, \mathbf{u}} \widehat{\mathcal{L}}(\mathbf{T}, \boldsymbol{\lambda}) + \frac{\beta}{2}\|\mathbf{u}\|_2^2 - \mathbf{u}^\top \boldsymbol{\lambda}.$$

Solving this regularized problem is equivalent to solving (4) via Algorithm (1) with a weight decay factor of $1 - \frac{1}{\beta}$ applied to the dual update; see Appendix A.5 for more details.

## 4 VIDEO DENOISING

We now consider a problem of video denoising, which is to reconstruct a sequence of signals $\mathbf{X} = [\mathbf{x}_0, \ldots, \mathbf{x}_T]$ from noisy measurements $\tilde{\mathbf{X}} = [\tilde{\mathbf{x}}_0, \ldots, \tilde{\mathbf{x}}_T]$, where $\tilde{\mathbf{x}}_t = \mathbf{x}_t + \boldsymbol{\epsilon}_t$, for all $t$, and $\boldsymbol{\epsilon}_t \sim \mathcal{N}(0, \sigma^2\mathbf{I})$. The goal is to minimize the reconstruction loss $f(\mathbf{X}, \Phi(\tilde{\mathbf{X}}; \mathbf{T})) = \frac{1}{T}\|\mathbf{X} - \mathbf{Y}\|_{\mathcal{F}}^2$, where $\mathbf{Y} = \Phi(\tilde{\mathbf{X}}; \mathbf{T})$ is the output sequence. Our experiment setup follows the denoising experiments in De Weerdt et al. (2023) and Luong et al. (2021). We train and evaluate our models on the CUHK Avenue Lu et al. (2013), UCSD Anomaly Detection Mahadevan et al. (2010) and ShanghaiTech Campus Luo et al. (2017) datasets. Refer to Appendix B for more details.

**In-distribution (ID) and out-of-distribution (OOD) evaluation.** In light of Theorem 4, which provides OOD generalization guarantees for models trained to satisfy descent constraints, we aim to contrast the performance of constrained and unconstrained models under distribution shifts. To accomplish this, we train denoisers with perturbation level $\gamma_{\text{train}}$, and evaluate on testing perturbations $\gamma \in [0.0, 1.0]$. We refer to the testing perturbation levels $\gamma \leq \gamma_{\text{train}}$ as ID, and to larger perturbations as OOD. Noisy signals $\tilde{\mathbf{X}}$ are generated with $\sigma = \gamma \cdot \sigma_x$, where $\sigma_x$ is the standard deviation of the clean data $\mathbf{X}$.

**Architectures.** We train three transformer architectures. The first is a standard pretrained Vision Transformer (ViT) (Dosovitskiy et al., 2021). The other two are symmetric transformers from the unrolling literature: Deep Unfolded Sequential Transformer (DUST) (De Weerdt et al., 2023) and Unrolled Transformer (UT) (Yang et al., 2022). All models follow the formulation in (1) and (2), except that in DUST and UT, the learnable parameters of the attention layers are tied, i.e., $\mathbf{Q}_l = \mathbf{K}_l = \mathbf{V}_l = \mathbf{D} \in \mathbb{R}^{D \times N}$, for all $l$. The key difference between DUST and UT lies in the choice of nonlinearities: DUST employs a soft-threshold nonlinearity, while UT uses ReLU. Notably, in DUST, the parameter $\mathbf{D}$ is interpreted as an overcomplete dictionary to create sparse reconstructions of video frames. Consequently, the dimensions satisfy $N < D$, marking a significant architectural deviation from conventional transformers, where attention matrices are low-rank projections. We defer further discussion about DUST and UT, along with additional implementation details, to Appendix C.

**Training.** We train each model under two settings: i) without constraints via the ERM formulation in (3), and ii) with descent constraints as specified in (4), employing a constant schedule $\alpha_l = \alpha$. To prevent the model from trivially satisfying the constraints by increasing the initial value of $f(\cdot)$, we include an additional constraint $f(\mathbf{X}, \Phi_1(\tilde{\mathbf{X}}; \mathbf{T})) \leq (1 - \alpha)f_0$, where $f_0$ is a fixed reference value. The unconstrained models are trained using ADAM, and the constrained ones are trained via Algorithm 1. For each dataset and each transformer architecture, we run experiments with different numbers of layers $L \in \{3, 5, 7\}$ and varying levels of perturbations. Here we present the results obtained for $\gamma_{\text{train}} = 0.13$ by selecting the best run across all values of $L$, and leave additional results for other training perturbations to Appendix B.4. All runs share the hyperparameters selected via a grid search, including the step size $\alpha$, the reference value $f_0$, the learning rates $\eta_1, \eta_2$, and the resilience coefficient $\beta$.

**Out-of-Distribution RMSE.** Figure 2 reports results at training perturbation level $\gamma_{\text{train}} = 0.13$, where the best $L$ for each model is selected based on validation RMSE. We observe that in five out of nine cases, constrained models outperform their unconstrained counterparts. Notably, on UCSD, all constrained models show a consistently lower RMSE score for values of $\gamma \geq 1$. In three cases, constrained models have comparable performance. Finally, in one case (UT-Avenue), OOD performance is slightly degraded.

**In-Distribution tradeoff.** The OOD curves of Figure 2 also allow us to analyze the tradeoff between ID and OOD performance. One noteworthy case is UT on ShanghaiTech, where constrained UT trades off a higher ID reconstruction error for lower error at higher perturbation levels, with an inflection point around $\gamma = 0.75$. In general, constrained models show slight to no degradation in RMSE relative to unconstrained ones. These results suggest that descent constraints improve robustness with little to no sacrifice in ID performance.

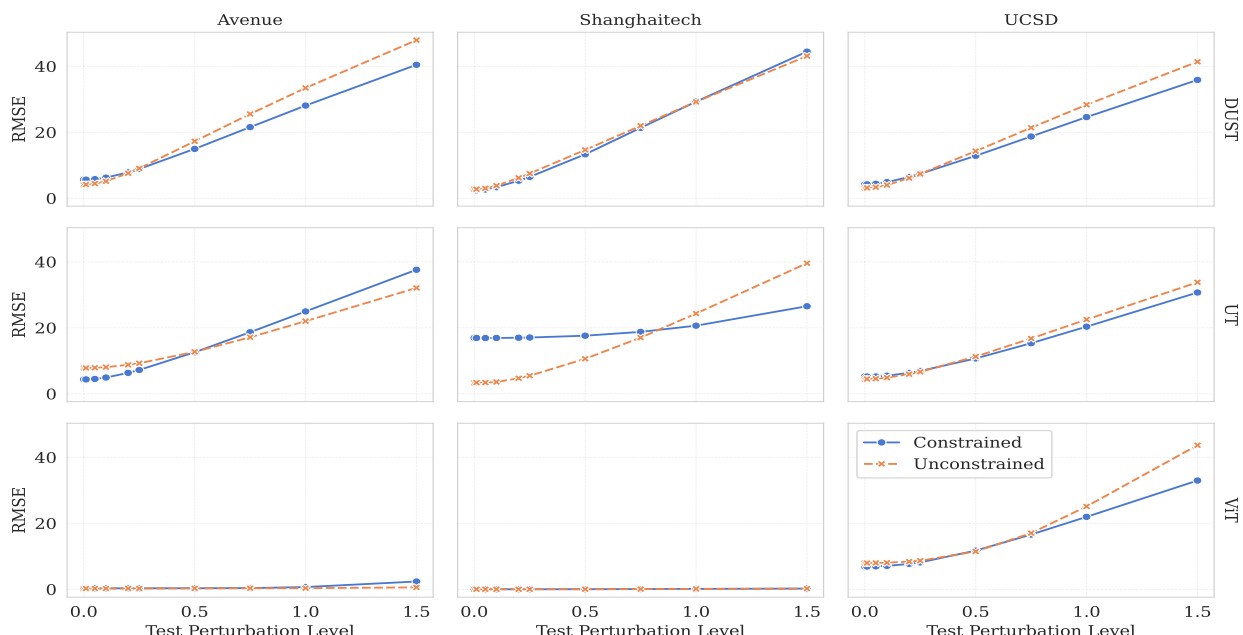

Figure 2: **Video denoising error vs. test perturbation** $\gamma$ (RMSE $\downarrow$, lower is better). Columns are datasets, rows are architectures. Solid lines are constrained models; dashed lines are unconstrained. Each plot shows RMSE over increasing test perturbation levels ($\gamma$). All models were trained with perturbation $\gamma_{\text{train}} = 0.13$.

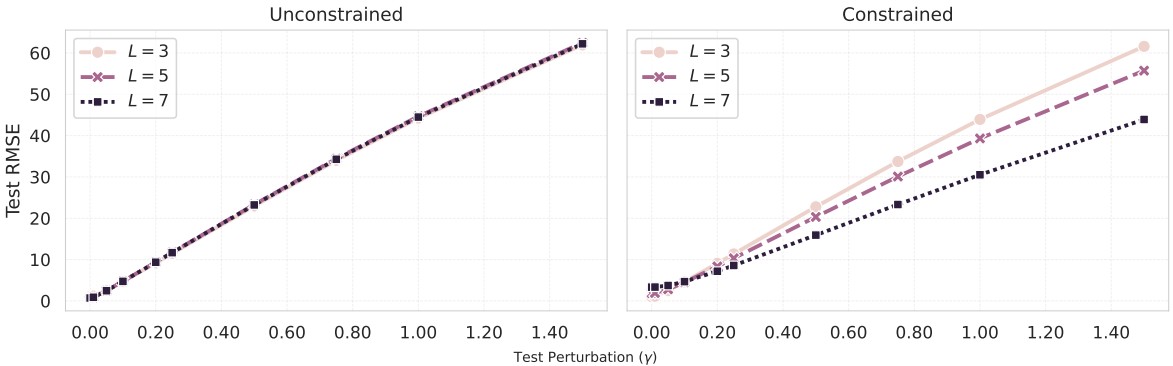

Figure 3: Effects of descent constraints on the OOD performance of (left, $\downarrow$ lower is better) unconstrained DUST and (right, $\downarrow$ lower is better) constrained DUST trained on UCSD with training $\gamma = 0$. Deeper constrained models show improved OOD performance, while increasing depth does not benefit unconstrained models.

**Increasing depth influences OOD performance** Finally, we show the effects of increasing the number of layers in Figure 3. As we train deeper models, we observe an decrease in RMSE for higher test $\gamma$, an effect that is not present in unconstrained models under the same settings. This is consistent with our theory: as the number of layers increases, the unrolled models converge to the optimal of the statistical loss under shifted distributions.

## 5 TEXT CLASSIFICATION WITH PERTURBED EMBEDDINGS

Our second use case is text classification in the presence of input noise. The task is to minimize negative cross-entropy, $f(\tilde{\mathbf{X}}, \mathbf{q}, \Phi(\tilde{\mathbf{X}}; \mathbf{T})) = -\sum_{c=1}^{C} q_c \log \psi(\Phi(\tilde{\mathbf{X}}; \mathbf{T}))_c$. Here, $\tilde{\mathbf{X}}$ are perturbed token embeddings as in the previous task, $\mathbf{q} \in \mathbb{R}^C$ are class labels and $\psi(\cdot)_c$ is the $c$-th component of a readout layer, $\psi(\cdot) : \mathbb{R}^N \to [0,1]^C$. We consider two different language understanding tasks: IMDb sentiment classification (Maas et al., 2011) and Multi-Genre Natural Language Inference (MNLI) from the GLUE benchmark (Wang et al., 2019).

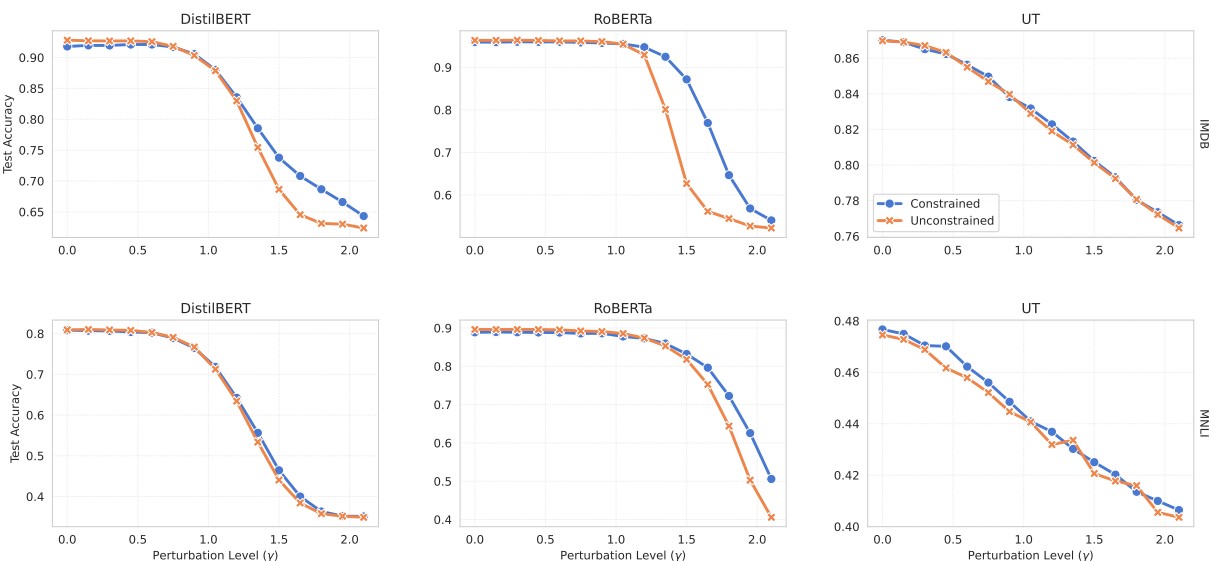

Figure 4: **Text classification accuracy vs. test perturbation** (Accuracy ↑, higher is better). Columns are datasets, rows are architectures. Solid lines denote constrained models; dashed lines denote unconstrained. Each plot shows accuracy over increasing test perturbation levels ($\gamma$). All models were trained with perturbation $\gamma_{\text{train}} = 0.8$.

**Architectures.** We adapt two pretrained language models, DistilBERT (Sanh et al., 2020) ($L = 12$) and RoBERTa (Liu et al., 2019b) ($L = 24$) and one unrolled transformer, UT (Yang et al., 2022). For MNLI, we omit RoBERTa due to the high computational cost. The architecture of the classifiers is identical to (1) and (2), except for the addition of a readout layer $\psi(\cdot)$, implemented as a single linear layer followed by a softmax nonlinearity. This readout is shared across all transformer layers to extract a label prediction from intermediate outputs' representations. The input to the readout is the [CLS] token for DistilBERT, and the average pooling of the output vectors of $\mathbf{Y}_l$ for unrolled transformer.

**Training.** As in video, we train unconstrained models with ERM, whereas constrained models are trained with our primal-dual algorithm under constant descent schedule, initialized from a reference value $f_0$. The hyperparameters are the same as specified above. In this case, constrained models proved more sensitive to dual hyperparameters, therefore we tuned each individually and report their best runs. For a fair comparison, we run the same number of unconstrained runs and report its best as well. The experiment setting is $\gamma_{\text{train}} = 0.8$, and $L \in \{3, 5, 7, 9\}$. We present summarized results on additional values of training perturbation in Appendix B.4.

**Out-of-Distribution Accuracy.** We observe a notable improvement in OOD robustness for DistilBERT and RoBERTa models on IMDb, as ilustrated in Figure 4. The gap between constrained and unconstrained is larger for RoBERTa, suggesting that the benefits of descent constraints are more significant with larger models. Constrained UT runs show comparable behavior to unconstrained runs. In general, the effects of constraints in MNLI are less pronounced.

**Preserved in-distribution accuracy.** In Figure 4, the accuracies at lower perturbation levels are the in-distribution performance. In all scenarios, we observe comparable results between constrained and unconstrained settings. In some cases, such as DistilBERT-IMDb, a marginal tradeoff between ID and OOD performance is observed.

**Differential Privacy.** This robust classifier is of interest in the context of local differential privacy. By degrading more smoothly to various levels of input perturbation, our model can support different degrees of privacy as required by a user at inference time, with less performance degradation than training only on perturbed inputs. A more thorough exploration of the application of our method to differential privacy is a promising future work direction.

### 5.1 ABLATION ANALYSIS

Our training method introduces two main hyperparameters: the layerwise step size $\alpha$, and the dual learning rate $\eta_2$. In Figure 5, we compare the ID and OOD performances for varying values of $\alpha$. Consistent with our theory, as $\alpha$ increases, we observe an improvement in OOD accuracy, and a less than 0.3% in distribution degradation. In Appendix B.6, we also provide experimental details and show that performance is consistent across different values of $\eta_2$.

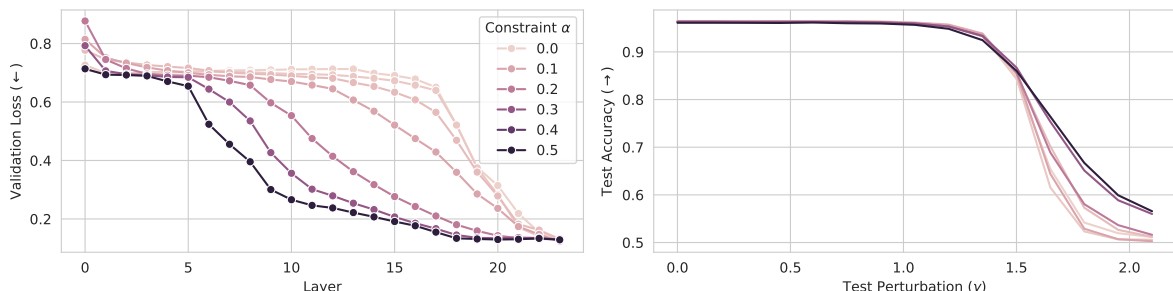

Figure 5: Ablation analysis of constrained RoBERTa on IMDb. Left: layerwise validation loss ($\downarrow$ lower is better), right: test accuracy across test perturbation levels $\uparrow$ higher is better. Increasing the constraint step size $\alpha$ induces monotonic descent along the layers, while improving OOD robustness.

## 6 UNROLLING ON LARGE SCALE MODELS

The pretrained models of Sections 4 and 5 suggest that the OOD robustness effect remains present with increased model size. To test whether this trend is persistent, we consider supervised fine-tuning on Llama 3.1 8B Grattafiori et al. (2024) on the Alpaca instruction dataset Taori et al. (2023). The setup is as in Section 5: the loss is cross-entropy, with $C$ being the vocabulary size, and we evaluate robustness by calculating mean token accuracy on a test set, with increasingly perturbed sequences. Figure 6 shows that the constrained model exhibits improved layerwise descent, more robustness to input perturbations, and virtually no ID degradation. For downstream task evaluation, we run AlpacaEval Taori et al. (2023) with a GPT-5 judge comparing the outputs of constrained and unconstrained models. For completions with input perturbations of $\gamma = 0.0$, the length-controlled win rate of constrained over unconstrained is 50%, indicating that in-distribution performance is preserved. With $\gamma = 1.5$, the length-controlled win rate is 69.93% favoring the constrained model, which confirms that the model retains its capabilities in the presence of input noise. More empirical exploration is necessary to understand the effects on downstream task performance. Appendix B.7 includes experimental details and completion examples.

## 7 CONCLUSIONS

This work presented a framework for training unrolled transformers by imposing descent constraints on the inter mediate outputs of the model. We developed a dual training algorithm for constrained transformers and showed that descent constraints ensure layerwise convergence to a near-optimal value of the statistical loss. We showed theoretically that our unrolled transformers retain their descent properties under distribution shifts, and that this enables improved OOD generalization. This was verified empirically with various transformer-based architectures in video denoising, language classification with perturbed embeddings and LLM supervised finetuning. In the case of language, this robust transformer can be applied to support varying levels of local Differential Privacy at inference time for a better utility tradeoff.

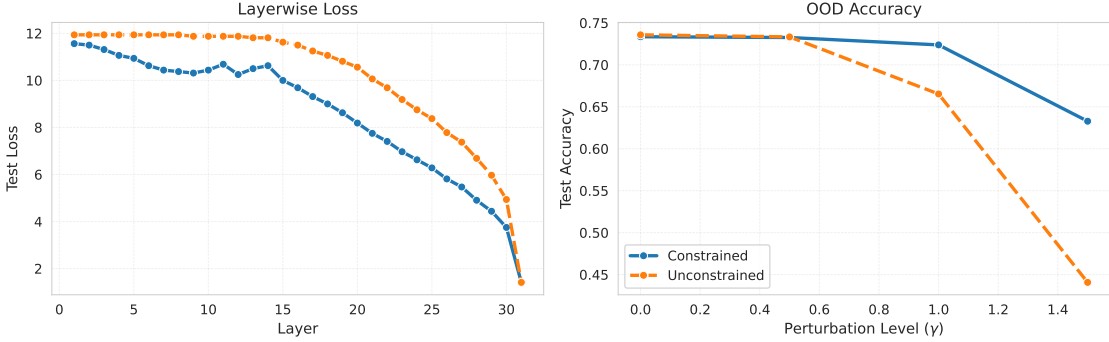

Figure 6: **Constrained Llama 8B.** Left: layerwise test loss ($\downarrow$ lower is better). Right: test token accuracy across perturbation levels ($\uparrow$ higher is better).

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

# A  MATHEMATICAL PROOFS

In this Appendix, we present the theoretical aspects of our approach, including proofs to our theorem and corollaries.

## A.1  CONSTRAINED LEARNING THEOREM

The Constrained Learning Theorem (CLT) characterizes the duality gap in constrained learning problems. Such problems are highly non-convex, and in general, nonconvex constrained problems lack guarantees of zero duality gap. However, CLT establishes that constrained learning problems exhibit small duality gap due to the high expressivity of neural networks.

**Theorem 1** (CLT (Chamon et al., 2023)). *Let $(\mathbf{T}^*, \boldsymbol{\lambda}^*)$ be a stationary point of* (6) *and $P^*$ denote the optimal value of the statistical loss function in* (4). *Under Assumptions 1 - 5, it holds, for some constant $\rho$, that*

$$|P^* - \widehat{D}^*| \leq C\nu + \rho\,\zeta(M, \delta), \quad and$$

$$\mathbb{E}\Big[f\big(\mathbf{X}, \Phi_l(\mathbf{X}; \mathbf{T}^*)\big)\Big] - (1 - \alpha_l)\,\mathbb{E}\Big[f\big(\mathbf{X}, \Phi_{l-1}(\mathbf{X}; \mathbf{T}^*)\big)\Big] \leq \zeta(M, \delta), \; \forall l,$$

*with probability $1 - \delta$ each, and with $\rho = \max\{\|\boldsymbol{\lambda}^*\|, \|\bar{\boldsymbol{\lambda}}^*\|\}$, where $\bar{\boldsymbol{\lambda}}^* = \operatorname{argmax}_{\boldsymbol{\lambda}} g(\boldsymbol{\lambda})$ is the optimal multiplier of the statistical dual function. Moreover, $\nu$ and $\zeta(M, \delta)$ are the expressivity parameter and the sample complexity, respectively, and $C$ is a Lipschitz constant.*

We refer the reader to (Chamon et al., 2023) for detailed proofs and discussions. In the following, we state the assumptions under which the theorem holds.

**Assumption 1.** *The loss function $f$ is $C$-Lipschitz continuous and bounded.*

**Assumption 2.** *Let $\Phi_l \in \mathcal{P}_l$ be a model with $l$ unrolling layers and parametrization $\mathbf{T}$. Denote the convex hull of $\mathcal{P}_l$ as $\bar{\mathcal{P}}_l := \overline{conv}(\mathcal{P}_l)$. Then, for every $\bar{\Phi} \in \bar{\mathcal{P}}_l$ and every $l$, there exist a parametrization $\mathbf{T}$ such that*

$$\mathbb{E}\Big[ \big\| \Phi_l(\mathbf{X}; \mathbf{T}) - \bar{\Phi}(\mathbf{X}) \big\|_{\mathcal{F}} \Big] \leq \nu, \tag{10}$$

*for any $\nu > 0$.*

**Assumption 3.** *Let $\mathcal{Y}$ denote the domain of the transformer's output. The set $\mathcal{Y}$ is either (i) finite, as in classification tasks, or (ii) compact, in which case the descent constraints are uniformly continuous with respect to the total variation topology for every $\bar{\Phi} \in \bar{\mathcal{P}}_L$. Moreover, the conditional distributions $\mathbf{X}|\mathbf{Y}$ are nonatomic.*

**Assumption 4.** *There exists $\zeta(M, \delta) \geq 0$ that is monotonically decreasing with the number of realizations $M$, for which it holds, for all $l$, with probability $1 - \delta$,*

$$\left| \mathbb{E}\big[f\big(\mathbf{X}, \Phi_l(\mathbf{X}; \mathbf{T})\big)\big] - \widehat{\mathbb{E}}\big[f\big(\mathbf{X}, \Phi_l(\mathbf{X}; \mathbf{T})\big)\big] \right| \leq \zeta(M, \delta), \tag{11}$$

*where $\widehat{\mathbb{E}}$ denotes the sample mean.*

**Assumption 5.** *There exists a parametrization of $L$ layers, $\Phi \in \mathcal{P}_L$, that is strictly feasible, i.e.,*

$$\mathbb{E}\Big[f\big(\mathbf{X}, \Phi_l(\mathbf{X}; \mathbf{T})\big)\Big] - (1 - \alpha_l)\,\mathbb{E}\Big[f\big(\mathbf{X}, \Phi_{l-1}(\mathbf{X}; \mathbf{T})\big)\Big] \leq -C\nu - \xi, \tag{12}$$

$$\widehat{\mathbb{E}}\Big[f\big(\mathbf{X}, \Phi_l(\mathbf{X}; \mathbf{T})\big)\Big] - (1 - \alpha_l)\,\widehat{\mathbb{E}}\Big[f\big(\mathbf{X}, \Phi_{l-1}(\mathbf{X}; \mathbf{T})\big)\Big] \leq -\xi, \tag{13}$$

*for all $l$, with $\xi > 0$.*

These assumptions are readily satisfied in practical settings. Assumption 1 induces Lipschitz continuity and holds for a wide class of loss functions, including $\ell_1$ and $\ell_2$ norms. Assumption 2 invokes the universal approximation theorem, which ensures that the parametrization is sufficiently rich to approximate any function $\bar{\Phi}$ up to a factor $\nu$. This property has been established for transformers in (Yun et al., 2020) and is the core reason that zero duality gap holds for constrained learning. Assumption 3 requires the transformer's output to be bounded, which can be guaranteed by restricting the input domain to a compact set and using bounded learnable parameterization. Additionally, it requires the conditional probabilities to be nonatomic. Assumption 4 imposes a mild assumption on the sample complexity, allowing us to replace statistical expectations with sample means. The strict feasibility condition in Assumption 5 can also be achieved by appropriately adjusting the design parameter $\alpha_l$ or using resilient constrained learning (Hounie et al., 2023).

## A.2 Proof of Theorem 2

Consider a probability space $(\Omega, \mathcal{F}, P)$, where $\Omega$ is a sample space, $\mathcal{F}$ is a sigma algebra, and $P : \mathcal{F} \to [0, 1]$ is a probability measure. We define a random variable $X : \Omega \to \mathbb{R}$ and write $P(\{\omega : X(\omega) = 0\})$ as $P(X = 0)$ to keep equations concise. We also define a filtration of $\mathcal{F}$ as $\{\mathcal{F}_l\}_{l>0}$, which can be thought of as an increasing sequence of $\sigma$-algebras with $\mathcal{F}_{l-1} \subset \mathcal{F}_l$. We assume that the outputs of the unrolled layers $\mathbf{Y}_l$ are adapted to $\mathcal{F}_l$, i.e., $\mathbf{Y}_l \in \mathcal{F}_l$, for all $l$.

A stochastic process $X_k$ is said to form a supermartingale if $\mathbb{E}[X_k | X_{k-1}, \ldots, X_0] \leq X_{k-1}$. This inequality implies that given the past history of the process, the future value $X_k$ is not, on average, larger than the latest one. In the following, we restate Theorem 2 before we provide a proof that uses a supermartingale argument similar to proofs of convergence of stochastic descent algorithms. We follow a similar line of reasoning to that in (Hadou et al., 2024b).

In our analysis, we consider a *functional* minimizer, $\phi^* : \mathbb{R}^{N \times T} \to \mathbb{R}^{N \times T}$, of the statistical loss:

$$\phi^* = \operatorname*{argmin}_{\phi} \mathbb{E}\big[ f(\mathbf{X}, \phi(\mathbf{X})) \big]. \tag{14}$$

We evaluate the optimality of our constrained unrolled transformers by comparing the statistical loss they achieve with the optimal value $\mathbb{E}\big[ f(\mathbf{X}, \phi^*(\mathbf{X})) \big]$. In Theorem 2, we argue that this difference is guaranteed to vanish asymptotically– in the number of layers–falling below a small threshold that depends on the sample complexity, the stepsize and the expressivity of the transformers. Theorem 2 holds under the following condition:

**Assumption 6.** *The loss function $f$ is $C$-Lipschitz continuous in its second argument, and there exists a parametrization $\mathbf{T}$ with $l$ layers that satisfies*

$$\mathbb{E}\big[ \|\Phi_l(\mathbf{X}; \mathbf{T}) - \phi(\mathbf{X})\|_{\mathcal{F}} \big] \leq \nu, \quad \forall l \tag{15}$$

*for some $\nu > 0$ and any $\phi : \mathbb{R}^{N \times T} \to \mathbb{R}^{N \times T}$.*

The Lipschitz continuity assumption is standard in the analysis of convergence. The second part of the assumption refers to the universal approximation of transformers, which has been established in (Yun et al., 2020). Under these assumptions, we can prove the convergence guarantees of the constrained unrolled transformers as follows.

**Theorem 2** (Convergence Guarantees)**.** *Given a constrained unrolled transformer $\mathbf{T}^*$, which satisfies Theorem 1, and a functional minimizer $\phi^*$ as in (32), whose output for a given input $\mathbf{X}$ is denoted by $\mathbf{Y}^* = \phi^*(\mathbf{X})$. Then, under Assumption 6, it holds that*

$$\lim_{l \to \infty} \min_{k \leq l} \mathbb{E}\Big[ f(\mathbf{X}, \Phi_k(\mathbf{X}; \mathbf{T}^*)) - f(\mathbf{X}, \mathbf{Y}^*) \Big] \leq \frac{1}{\alpha} \left( \zeta(M, \delta) + \frac{C\delta\nu}{1-\delta} \right), \quad a.s.$$

*with $\alpha_l = \alpha$, for all $l$.*

*Proof.* Let $A_l \in \mathcal{F}_l$ be the event that the descent constraint in (4) at layer $l$ is satisfied, and denote the output of layer $l$, $\Phi_l(\mathbf{X}; \mathbf{T}^*)$, as $\mathbf{Y}_k$. By the total expectation theorem, we have

$$\begin{aligned} \mathbb{E}\big[ f(\mathbf{X}, \mathbf{Y}_l) - f(\mathbf{X}, \mathbf{Y}^*) \big] \\ = P(A_l)\mathbb{E}\big[ f(\mathbf{X}, \mathbf{Y}_l) - f(\mathbf{X}, \mathbf{Y}^*) | A_l \big] + P(A_l^c)\mathbb{E}\big[ f(\mathbf{X}, \mathbf{Y}_l) - f(\mathbf{X}, \mathbf{Y}^*) | A_l^c \big], \end{aligned} \tag{16}$$

with $P(A_l) = 1 - \delta$. The first term on the right-hand side is the expectation conditioned on the descent constraint being met, which is bounded above according to Theorem 1. The second term represents the complementary event $A_l^c \in \mathcal{F}_l$, and is also bounded above:

$$\begin{aligned} \mathbb{E}\big[ f(\mathbf{X}, \mathbf{Y}_l) - f(\mathbf{X}, \mathbf{Y}^*) \big] &= \mathbb{E}\big[ |f(\mathbf{X}, \mathbf{Y}_l) - f(\mathbf{X}, \mathbf{Y}^*)| \big] \\ &\leq C\,\mathbb{E}\big[ \|\mathbf{Y}_k - \mathbf{Y}^*\|_{\mathcal{F}} \big] \\ &= C\,\mathbb{E}\big[ \|\Phi_k(\mathbf{X}; \mathbf{T}^*) - \phi^*(\mathbf{X})\|_{\mathcal{F}} \big] \leq C\nu, \end{aligned} \tag{17}$$

where $\|\cdot\|_{\mathcal{F}}$ is the Frobenius norm. The first equality is true since $f(\mathbf{X}, \mathbf{Y}^*) \leq f(\mathbf{X}, \mathbf{Y})$ by definition. The two inequalities follow from Assumption 6: the first inequality is a direct application of the Lipschitz continuity and the second one of the universal approximation property. Thus,

$$\begin{aligned} \mathbb{E}\big[ f(\mathbf{X}, \mathbf{Y}_l) - f(\mathbf{X}, \mathbf{Y}^*) \big] &\leq (1-\delta)(1-\alpha)\,\mathbb{E}\big[ f(\mathbf{X}, \mathbf{Y}_{l-1}) - f(\mathbf{X}, \mathbf{Y}^*) \big] \\ &\quad + (1-\delta)\zeta(M, \delta) + C\delta\nu, \end{aligned} \tag{18}$$

almost surely. We let $Z_l = \mathbb{E}\big[f(\mathbf{X}, \mathbf{Y}_l) - f(\mathbf{X}, \mathbf{Y}^*)\big]$, a random variable with a degenerate distribution, and $\eta = \frac{1}{\alpha}\left(\zeta(M, \delta) + \frac{C\delta\nu}{1-\delta}\right)$. We then convert (18) a supermartingale inequality:

$$
\begin{aligned}
\mathbb{E}\big[Z_l \mid \mathcal{F}_{l-1}\big] &\leq (1-\delta)(1-\alpha)\, Z_{l-1} + (1-\delta)\zeta(M, \delta) + C\delta\nu \\
&= (1-\delta)\, Z_{l-1} - (1-\delta)\Big(\alpha Z_{l-1} - \zeta(M, \delta) - \frac{C\delta\nu}{1-\delta}\Big) \\
&= (1-\delta)\, Z_{l-1} - (1-\delta)\Big(\alpha Z_{l-1} - \alpha\eta\Big).
\end{aligned}
\tag{19}
$$

The goal of the rest of the proof is to show that with growing $l$, $Z_l$ almost surely and infinitely often achieves values less that $\eta$, i.e.,

$$
\lim_{l\to\infty} \min_{k\leq l}\{Z_k\} \leq \eta \quad a.s. \tag{20}
$$

Equation (20) restates (7) using simplified notation. To this end, we introduce two auxiliary sequences:

$$
\begin{aligned}
\beta_l &:= Z_l \cdot \mathbf{1}\{Z_l^{\text{best}} > \eta\}, \\
\gamma_l &:= \alpha\,(Z_l - \eta) \cdot \mathbf{1}\{Z_l^{\text{best}} > \eta\},
\end{aligned}
\tag{21}
$$

where $Z_l^{\text{best}} = \min_{k\leq l}\{Z_k\}$ tracks the best-so-far value observed up to step $l$, and $\mathbf{1}\{.\}$ is an indicator function. Since $\eta$ is nonnegative, it follows that $\beta_l \geq 0$ and $\gamma_l \geq 0$, for all $l$.

The sequence $\beta_l$ mirrors the values of $Z_l$ while the best-so-far value $Z_l^{\text{best}}$ remains above the threshold $\eta$. Once $Z_l^{\text{best}}$ falls below $\eta$, the indicator function becomes zero and $\beta_l$ remains zero for all subsequent steps. Similarly, the sequence $\gamma_l$ holds the values of $\alpha(Z_l - \eta)$ only as long as $Z_l^{\text{best}}$ is above $\eta$, and also vanishes thereafter.

We now invoke the supermartingale convergence theorem (Robbins and Siegmund, 1971, Theorem 1) to show that $\beta_l$ converges almost surely and the sequence $\gamma_l$ is summable, which will facilitate the proof of (20). To apply this theorem, we first need to verify that the sequence $\beta_l$ forms a supermartingale.

A sequence $\beta_l$ is a supermartingale if the conditional expectation given the past is upper bounded by the most recent value, i.e., $\mathbb{E}[\beta_l \mid \mathcal{F}_{l-1}] \leq \beta_{l-1}$. The conditional expectation can be written as

$$
\mathbb{E}\big[\beta_l \mid \mathcal{F}_{l-1}\big] = \mathbb{E}\big[\beta_l \mid \mathcal{F}_{l-1}, \beta_{l-1} = 0\big]\,P(\beta_{l-1} = 0) + \mathbb{E}\big[\beta_l \mid \mathcal{F}_{l-1}, \beta_{l-1} \neq 0\big]\,P(\beta_{l-1} \neq 0), \tag{22}
$$

splitting the expectation into two cases: $\beta_{l-1} = 0$ and $\beta_{l-1} \neq 0$. When $\beta_{l-1} = 0$, (21) implies that the indicator function is zero and $Z_{l-1}^{\text{best}} \leq \eta$. In turn, $\beta_k = 0$ and $\gamma_k = 0$, for all $k \geq l-1$. Hence, the first term in (22) is zero,

$$
\mathbb{E}\big[\beta_l \mid \mathcal{F}_{l-1}, \beta_{l-1} = 0\big] = (1-\delta)(\beta_{l-1} - \gamma_{l-1}) = 0. \tag{23}
$$

When $\beta_{l-1} \neq 0$, the conditional expectation follows from the definition in (21),

$$
\begin{aligned}
\mathbb{E}\big[\beta_l \mid \mathcal{F}_{l-1}, \beta_{l-1} \neq 0\big] &= \mathbb{E}\big[Z_l \cdot \mathbf{1}\{Z_l^{\text{best}} > \eta\} \mid \mathcal{F}_{l-1}, \beta_{l-1} \neq 0\big] \\
&\leq \mathbb{E}\big[Z_l \mid \mathcal{F}_{l-1}, \beta_{l-1} \neq 0\big] \\
&\leq (1-\delta)\, Z_{l-1} - (1-\delta)\Big(\alpha Z_{l-1} - \alpha\eta\Big) \\
&= (1-\delta)(\beta_{l-1} - \gamma_{l-1}).
\end{aligned}
\tag{24}
$$

In the first equality, we plugin (21). The first inequality holds because the indicator function is either zero or one and the second inequality is a direct application of (19). The last equality results from that fact that the indicator function $\mathbf{1}\{Z_l^{\text{best}} > \eta\}$ is one since $\beta_{l-1} \neq 0$, which implies that $\beta_{l-1} = Z_{l-1}$ and $\gamma_{l-1} = \alpha(Z_{l-1} - \eta)$. Combining the results of (23) and (24), we find that

$$
\begin{aligned}
\mathbb{E}\big[\beta_l \mid \mathcal{F}_{l-1}\big] &\leq (1-\delta)(\beta_{l-1} - \gamma_{l-1})\Big[P(\beta_{l-1} = 0) + P(\beta_{l-1} \neq 0)\Big] \\
&= (1-\delta)(\beta_{l-1} - \gamma_{l-1}).
\end{aligned}
\tag{25}
$$

Hence, $\beta_l$ forms a supermartingale. By the supermartingale convergence theorem, (25) implies that (i) $\beta_l$ converges almost surely, and (ii) $\sum_{l=1}^{\infty} \gamma_l$ is almost surely summable (i.e., finite). When the latter is written explicitly, we get

$$
\sum_{l=1}^{\infty}\Big(\alpha Z_l - \alpha\eta\Big) \cdot \mathbf{1}\{Z_l^{\text{best}} > \eta\} < \infty, \quad a.s., \tag{26}
$$

Since $\gamma_l \geq 0$, for all $l$, (26) implies that the limit inferior and limit superior collapse to zero,

$$\liminf_{l \to \infty} \left( \alpha Z_l - \alpha \eta \right) \cdot \mathbf{1}\{Z_l^{\text{best}} > \eta\} = 0, \quad a.s. \tag{27}$$

Equation (27) is true if either there exist a sufficiently large $l$ such that $Z_l^{\text{best}} \leq \eta$ to set the indicator to zero or it holds that

$$\liminf_{l \to \infty} \left( \alpha Z_l - \alpha \eta \right) = 0, \quad a.s. \tag{28}$$

which is equivalent to having $\sup_l \inf_{m \geq l} Z_m = \eta$. Hence, there exists some large $l$ where $Z_l^{\text{best}} \leq \sup_l \inf_{m \geq l} Z_m$, which leads to the same upper bound. This proves the correctness of (20) and completes the proof. $\square$

## A.3 PROOF OF COROLLARY 3

The proof of Corollary 3 is adapted from (Hadou et al., 2024b) and is included here for completeness. The corollary holds under the following assumption:

**Assumption 7.** *There exists a non-negative asymmetric distance $d(\cdot, \cdot)$ between the input distribution $D_x$ and the OOD distribution $D_x'$ such that*

$$\mathbb{E}_{D_x}\left[ f\left( \mathbf{X}, \Phi_l(\mathbf{X}; \mathbf{T}^*) \right) \right] - \mathbb{E}_{D_x'}\left[ f\left( \mathbf{X}, \Phi_l(\mathbf{X}; \mathbf{T}^*) \right) \right] \leq C d(D_x, D_x')$$

*uniformly over the second argument with $C$ being a Lipschitz constant.*

**Corollary 3.** *Let $\mathbf{T}^*$ be a constrained unrolled transformer trained on a data distribution $D_x$. Then, for any shifted distribution $D_{x'}$ that satisfies Assumption 7, it holds with probability $1 - \delta$, for all $l$:*

$$\mathbb{E}_{D_x'}\left[ f\left( \mathbf{X}, \Phi_l(\mathbf{X}; \mathbf{T}^*) \right) \right] - (1 - \alpha_l)\, \mathbb{E}_{D_{x'}}\left[ f\left( \mathbf{X}, \Phi_{l-1}(\mathbf{X}; \mathbf{T}^*) \right) \right] \leq \zeta(M, \delta) + C\tau, \tag{29}$$

*where $\tau = d(D_x, D_{x'}) + d(D_{x'}, D_x)$, and $d(\cdot, \cdot)$ is a bounded asymmetric distance metric.*

*Proof.* We start by adding and subtracting the following two quantities $\mathbb{E}_{D_x}\left[ f\left( \mathbf{X}, \Phi_l(\mathbf{X}; \mathbf{T}^*) \right) \right]$ and $(1 - \epsilon)\mathbb{E}_{D_x}\left[ \|\nabla f(\mathbf{y}_{l-1}; \mathbf{x})\|_2 \right]$ from the quantity we seek to evaluate, i.e., we get

$$\begin{aligned}
\mathbb{E}_{D_x'}&\left[ f\left( \mathbf{X}, \Phi_l(\mathbf{X}; \mathbf{T}^*) \right) \right] - (1 - \alpha_l)\, \mathbb{E}_{D_{x'}}\left[ f\left( \mathbf{X}, \Phi_{l-1}(\mathbf{X}; \mathbf{T}^*) \right) \right] \\
&= \mathbb{E}_{D_x'}\left[ f\left( \mathbf{X}, \Phi_l(\mathbf{X}; \mathbf{T}^*) \right) \right] - \mathbb{E}_{D_x}\left[ f\left( \mathbf{X}, \Phi_l(\mathbf{X}; \mathbf{T}^*) \right) \right] \\
&\quad + (1 - \alpha_l)\Big[ \mathbb{E}_{D_x}\left[ f\left( \mathbf{X}, \Phi_{l-1}(\mathbf{X}; \mathbf{T}^*) \right) \right] - \mathbb{E}_{D_{x'}}\left[ f\left( \mathbf{X}, \Phi_{l-1}(\mathbf{X}; \mathbf{T}^*) \right) \right] \Big] \\
&\quad + \mathbb{E}_{D_x}\left[ f\left( \mathbf{X}, \Phi_l(\mathbf{X}; \mathbf{T}^*) \right) \right] - (1 - \alpha_l)\, \mathbb{E}_{D_x}\left[ f\left( \mathbf{X}, \Phi_{l-1}(\mathbf{X}; \mathbf{T}^*) \right) \right].
\end{aligned} \tag{30}$$

The right-hand side consists of three terms that can be bounded above with positive quantities according to Assumption 7 and Theorem 1. Therefore, the descent constraints under the new distribution $D_{x'}$ can be bounded above by

$$\begin{aligned}
\mathbb{E}_{D_x'}\left[ f\left( \mathbf{X}, \Phi_l(\mathbf{X}; \mathbf{T}^*) \right) \right] &- (1 - \alpha_l)\, \mathbb{E}_{D_{x'}}\left[ f\left( \mathbf{X}, \Phi_{l-1}(\mathbf{X}; \mathbf{T}^*) \right) \right] \\
&\leq C d(D_x', D_x) + C(1 - \alpha_l) d(D_x, D_x') + \zeta(M, \delta) \\
&\leq C d(D_x', D_x) + C d(D_x, D_x') + \zeta(M, \delta).
\end{aligned} \tag{31}$$

Notice that this inequality holds with probability $1 - \delta$ since the upper bound in Theorem 1 also holds with the same probability. This completes the proof. $\square$

## A.4 PROOF OF COROLLARY 4

We evaluate the OOD generalizability of the constrained unrolled transformers by comparing their performance to that of a functional minimizer of the statistical loss under the shifted distribution, i.e.,

$$\widehat{\phi}^* = \underset{\phi}{\operatorname{argmin}}\ \mathbb{E}_{D_{x'}}\left[ f\left( \mathbf{X}, \phi(\mathbf{X}) \right) \right], \tag{32}$$

where $\phi : \mathbb{R}^{N \times T} \to \mathbb{R}^{N \times T}$ maps $\mathbf{X}$ to $\mathbf{Y}$.

**Corollary 4** (Out-of-Distribution Generalization). *Let $\widehat{\phi}^*$ be a functional minimizer of the statistical loss evaluated on $D_{x'}$ and map input $\mathbf{X}$ to an estimation $\widehat{\mathbf{Y}}^*$. Then, the constrained unrolled transformer trained on $D_x$ satisfies*

$$\lim_{l \to \infty} \min_{k \leq l}\ \mathbb{E}_{D_{x'}}\left[ f\left( \mathbf{X}, \Phi_k(\mathbf{X}; \mathbf{T}^*) \right) - f\left( \mathbf{X}, \widehat{\mathbf{Y}}^* \right) \right] \leq \frac{1}{\alpha}\left( \zeta(M, \delta) + C\tau + \frac{C\delta\nu}{1 - \delta} \right). \tag{33}$$

*Proof.* The proof of this corollary proceeds identically to that of Theorem 2 (see Appendix A.2), except it is initialized with the inequality in Corollary 3. $\square$

## A.5 RESILIENT CONSTRAINED LEARNING

Resilient constrained learning (Hounie et al., 2023) aims to find an optimal relaxation of the constraints to ensure the feasibility of the learning problem. to this end, it introduces a slack variable $\mathbf{u} \in \mathbb{R}^L_+$ and reformulates the constrained training problem in (4) as

$$
\mathbf{T}^* = \underset{\mathbf{T}, \mathbf{u}}{\operatorname{argmin}} \quad \mathbb{E}\Big[ f\big( \mathbf{X}, \Phi(\mathbf{X}; \mathbf{T}) \big) \Big] + h(\mathbf{u}),
$$
$$
\text{subject to } \mathbb{E}\Big[ f\big( \mathbf{X}, \Phi_l(\mathbf{X}; \mathbf{T}) \big) \Big] \leq (1 - \alpha_l) \mathbb{E}\Big[ f\big( \mathbf{X}, \Phi_{l-1}(\mathbf{X}; \mathbf{T}) \big) \Big] + u_l, \quad \forall l, \tag{34}
$$

where $h(\cdot)$ is a convex relaxation cost, e.g., an $\ell_2$ norm. Similarly to (4), we tackle (34) in the dual domain by defining the corresponding Lagrangian function as

$$
\widehat{\mathcal{L}}_R(\mathbf{T}, \boldsymbol{\lambda}, \mathbf{u}) = \widehat{\mathcal{L}}(\mathbf{T}, \boldsymbol{\lambda}) + \frac{\beta}{2}\|\mathbf{u}\|_2^2 - \mathbf{u}^\top \boldsymbol{\lambda}, \tag{35}
$$

where $\widehat{\mathcal{L}}$ is the Lagrangian of the original problem. In (35), we choose the cost function $h$ to be the $\ell_2$ norm, i.e., $h(\mathbf{u}) = \frac{\beta}{2}\|\mathbf{u}\|_2^2$. The associated dual problem becomes

$$
\widehat{D}_R^* = \max_{\boldsymbol{\lambda}} \min_{\mathbf{T}, \mathbf{u}} \widehat{\mathcal{L}}(\mathbf{T}, \boldsymbol{\lambda}) + \frac{\beta}{2}\|\mathbf{u}\|_2^2 - \mathbf{u}^\top \boldsymbol{\lambda}. \tag{36}
$$

The optimal slack variable can be obtained by taking the derivative of the objective $\widehat{\mathcal{L}}_R$ and equating it to zero. This results in $\mathbf{u}^*(\boldsymbol{\lambda}) = \frac{1}{\beta}\boldsymbol{\lambda}$, and, in turn, the dual problem reduces to

$$
\widehat{D}_R^* = \max_{\boldsymbol{\lambda}} \min_{\mathbf{T}} \widehat{\mathcal{L}}(\mathbf{T}, \boldsymbol{\lambda}) - \frac{1}{2\beta}\|\boldsymbol{\lambda}\|_2^2. \tag{37}
$$

Problem (37) is a regularized variant of the empirical dual problem in (6), and can be solved with the same optimization scheme: alternating between minimizing with respect to $\mathbf{T}$ and maximizing over $\boldsymbol{\lambda}$. The gradient with respect to $\mathbf{T}$ remains unchanged, resulting in the same primal update as in Algorithm 1. However, the gradient with respect to $\boldsymbol{\lambda}$ now includes a regularized term, $-\frac{1}{\beta}\boldsymbol{\lambda}$, modifying the dual update to

$$
\boldsymbol{\lambda} = \left[ \left(1 - \frac{1}{\beta}\right) \boldsymbol{\lambda} + \eta_2 \nabla_{\boldsymbol{\lambda}} \widehat{\mathcal{L}}(\mathbf{T}, \boldsymbol{\lambda}) \right]_+. \tag{38}
$$

This formulation is analogous to applying weight decay in updating the Lagrangian multipliers and serves to stabilize their growth. In our experiments, we employ either the update rule in (38) or directly solve (36) via automatic differentiation, depending on the problem setting.

# B Experimental details

## B.1 Common Implementation details for Sections 4 and 5

**Training setting.** The goal of both experiments is to compare the behavior of constrained and unconstrained models under different settings. One setting is comprised of a model, a dataset, a perturbation level $\gamma$, and a depth $L$.

**Hyperparameters.** The common hyperparameters to tune are $\beta$, $\eta_1$, $\eta_2$, $\alpha$ and $f_0$. The hyperparameter search method differs in video and language, detailed in the next sections.

**Optimizers.** Unconstrained training uses one ADAM optimizer. Constrained training uses two ADAM optimizers, one for the neural network parameters and another one for optimizing the dual variables. We implement resilient constrained learning in video as presented in the original work (Hounie et al., 2023). For the language experiment, we use the weight decay formulation of resilience. However, as was noted in Section 3, both formulations are equivalent.

**Compute platform.** The video experiments were distributed between three machines: one machine has a single NVIDIA GeForce RTX 3080 Ti GPU, two of the machines have two NVIDIA GeForce RTX 3090 cards. The language experiments were run exclusively in the machines with two GPUs.

**Relevant libraries.** All of our experiments are implemented using PyTorch, version 2.6 for video and 2.7 for language. Additionally, the language experiment uses HuggingFace Datasets and Pytorch Lightning.

## B.2 Video Denoising Implementation Details

**Models.** We considered three models: UT, DUST, and ViT. While we generally follow the implementation from (De Weerdt et al., 2023), we make some minor simplifications to DUST and UT, which may make our results not directly comparable to theirs. These changes are explained in Appendix C To adapt ViT to the denoising task, we discard the classifier head, directly take each layer's output and interpret it as a reconstruction. It is worth noting that ViT processes each frame separately, while DUST and UT are natively designed to process sequences of patches. However, we reiterate that the goal of our experiments is *not* to compare performance across models, but rather contrast the constrained and unconstrained versions of each. Therefore, this difference is not significant for our purposes.

**Splits and data processing.** We reuse the preprocessing from (Luong et al., 2021), which consists of grayscaling, resizing to 160x160, and creating 16x16 patches. Vision Transformer uses its own out-of-the box processor on each frame.

**Initialization.** Dual variables and resilience slacks are initialized to zero. Model weight initialization in video uses a Discrete Cosine Transform for DUST and UT (Luong et al., 2021). ViT is initialized to pretrained weights.

**Metrics.** For a set of ground-truth images $\{\mathbf{Y}_i\}_{i=1}^N$ and their reconstructions $\{\widehat{\mathbf{Y}}_i\}_{i=1}^N$, the root mean squared error (RMSE) is defined as $\text{RMSE} = \sqrt{\frac{1}{N}\sum_{i=1}^N \|\widehat{\mathbf{Y}}_i - \mathbf{Y}_i\|_2^2}$. At test time, we evaluate RMSE under different perturbation levels, $\gamma \in \{0.01, 0.05, 0.1, 0.2, 0.25, 0.5, 0.75, 1.0, 1.5\}$. In the forthcoming extended results, we summarize model performance across different distribution shifts, we report the mean RMSE across perturbation levels.

**Primal warmup.** We train for an epoch without activating constraints as we empirically observed this aids with the stability of constrained training

**Resilience restarts.** After every epoch, we clamp the resilience slacks to zero. We empirically observe that initial relaxations tend to be high and then converge slowly. Restarting the slacks after every epoch helps converge to tighter feasible solutions more quickly.

**Hyperparameter tuning.** We perform a single hyperparameter search for each model with training perturbation $\gamma = 0.15$ and reuse the results for every setting. We fix $\eta_1 = 3 \times 10^{-4}$, except for runs of DUST-Avenue with $L = 9$, which use $\eta_1 = 8 \times 10^{-6}$. Table 1 shows the results of the hyperparameter search.

Table 1: Dual hyperparameters used for each model in the video denoising task.

| Model | $\alpha$ | $f_0$ | $\beta$ | $\eta_2$ |
|---|---|---|---|---|
| DUST | 6.50 | $3.10 \times 10^6$ | 0.75 | $2.78 \times 10^{-4}$ |
| UT | 5.50 | $5.30 \times 10^5$ | 0.78 | $3.20 \times 10^{-4}$ |
| ViT | 0.44 | $2.02 \times 10^2$ | 0.71 | $3.00 \times 10^{-4}$ |

Table 2: Dual hyperparameters for the best run of each model in the language task.

| Model | Dataset | $L$ | $\gamma$ | $\alpha$ | $f_0$ | $\beta$ | $\eta_2$ |
|---|---|---|---|---|---|---|---|
| DistilBERT | IMDB | 12 | 1.00 | 0.774 | 1.00 | 1.07 | $3.83 \times 10^{-2}$ |
| | MNLI | 12 | 1.00 | 0.774 | 1.00 | 1.99 | $1.51 \times 10^{-2}$ |
| UT | IMDB | 3 | 0.80 | 0.900 | 0.90 | 3.45 | $2.80 \times 10^{-2}$ |
| | MNLI | 3 | 0.00 | 0.900 | 0.90 | 1.98 | $9.12 \times 10^{-2}$ |

## B.3 LANGUAGE EXPERIMENT SETUP.

**Splits and data processing.** We rely on the standard train and test splits for each dataset. For the case of MNLI, there is a single step of preprocessing where we combine the premise and hypothesis into a single instance.

**Models.** We considered two models, a pretrained DistilBERT and UT. UT takes as input the same word embeddings as DistilBERT.

**Initialization.** Dual variables and resilience slacks are initialized to zero. UT uses Xavier initialization. DistilBERT is initialized to pretrained weights.

**Metric.** In language, we report the prediction accuracy, $\text{Acc}(\mathbf{x}, \mathbf{y}) = \frac{1}{M} \sum_{i=1}^{M} \mathbf{1}\{x_i = y_i\}$, where $M$ is the number of samples, $\mathbf{x}$ is the true vector of classes, and $\mathbf{y}$ is the predicted classes. To summarize, we estimate the AUC of the accuracies at different perturbation levels.

**Hyperparameter tuning.** For constrained training, we performed a Bayesian hyperparameter search with five runs per experimental setting. To maintain a fair comparison, unconstrained training was executed five times with different seeds, and we report the best result. This choice was motivated by observing a higher sensitivity of constrained experiments to dual hyperparameters compared to the video experiments. For brevity, Table 2 only lists hyperparameters corresponding to the best-performing run for each combination of dataset, model, and number of layers. In all settings, we fixed $\eta_1 = 10^{-5}$.

## B.4 EXTENDED RESULTS FOR VIDEO AND LANGUAGE

**Constraints Improve OOD Performance Across Settings.** In Section 4 and 5 we analyzed ID and OOD distribution for runs with a particular training perturbation. In Tables 3 and 4, we provide a summary of the complete suite of experiments for language and video experiments respectively. Note that RoBERTa-MNLI were omitted due to computational limitations. We can appreciate that in most settings, training with descent constraints increases performance when compared to unconstrained runs with the same settings. In many cases, these differences are significant. For instance, constrained DUST on the Avenue dataset with $L = 5$ and $\gamma = 0.15$, has an average RMSE of 14.72, while unconstrained is 28.269, a 71% reduction. On the language side, we find a similar result: constraints either improve or attain comparable AUC when compared across settings.

We also observe that a small number of constrained runs have very high RMSE, such as DUST with 7 layers on the ShanghaiTech dataset results in an RMSE above 131. The reason for these anomalies is challenges with primal-dual convergence. These results highlight the importance of carefully choosing the dual parameters and verifying training finalizes at a feasible solution.

**Monotonic Descent Behavior.** In Figure 7, we observe that constrained DUST trained on the UCSD dataset exhibits a monotonically decreasing loss $f$ across the layers. This effect persists consistently for models with different depths. In contrast, the descent behavior is absent in the unconstrained counterparts. Similar patterns were observed for other models and datasets. We note that due to the choice of the reference value $f_0$, the initial energy in some constrained runs may be higher than that of the unconstrained ones.

**Different Tradeoffs in Video Denoising.** Figure 8 shows three more examples where, with the same settings, constrained models achieve a better tradeoff between in-distribution and OOD reconstruction quality. In two of them (UT and ViT), we see constrained models trading off reconstruction quality at low levels of noise for better OOD performance. In the case of constrained DUST, however, we see a higher reconstruction quality compared to unconstrained in low-perturbation settings, and this difference gradually decreases with more test noise.

**Non-uniform Effect of Perturbation.** Figure 10 shows constrained and unconstrained OOD accuracy curves for UT and DistilBERT. In DistilBERT, as the perturbation level increases, the OOD accuracy of the constrained model slightly

Table 3: Average test RMSE over perturbations: Constrained vs Unconstrained (all settings)

| | $L$ | $\gamma_{\text{train}}$ | Avenue | | Shanghaitech | | UCSD | |
|---|---|---|---|---|---|---|---|---|
| | | | Constr | Unconstr. | Constr | Unconstr. | Constr | Unconstr. |
| **DUST** | 3 | 0.00 | 21.145 | **19.679** | **19.290** | 19.357 | **19.165** | 19.323 |
| | | 0.09 | **16.976** | 17.139 | 16.175 | **15.952** | **14.856** | 14.974 |
| | | 0.11 | **16.054** | 16.410 | **15.377** | 15.444 | **14.166** | 14.167 |
| | | 0.13 | **15.409** | 15.878 | 17.452 | **14.838** | 13.562 | **13.505** |
| | | 0.15 | **14.765** | 15.264 | **13.781** | 14.276 | 12.970 | **12.934** |
| | 5 | 0.00 | **17.650** | 21.072 | **18.643** | 19.174 | **17.517** | 19.469 |
| | | 0.09 | **15.599** | 20.344 | **15.018** | 15.259 | **14.362** | 14.443 |
| | | 0.11 | **15.848** | 20.960 | 13.633 | **12.463** | 105.767 | **20.965** |
| | | 0.13 | **18.057** | 22.368 | **13.231** | 13.738 | **13.240** | 13.257 |
| | | 0.15 | **14.724** | 28.269 | **12.410** | 18.636 | **12.642** | 12.840 |
| | 7 | 0.00 | **16.179** | 19.370 | 131.102 | **19.233** | **14.448** | 19.396 |
| | | 0.09 | **28.916** | 36.374 | 132.968 | **14.754** | **13.040** | 25.968 |
| | | 0.11 | **14.885** | 25.883 | 138.362 | **13.470** | 22.551 | **20.503** |
| | | 0.13 | **14.515** | 63.309 | **13.106** | 13.502 | **12.365** | 62.931 |
| | | 0.15 | **14.330** | 30.617 | **11.986** | 13.065 | **12.024** | 23.470 |
| **UT** | 3 | 0.00 | 16.770 | **16.180** | **17.691** | 18.471 | **15.074** | 15.353 |
| | | 0.09 | **14.098** | 14.639 | **14.381** | 15.026 | **12.938** | 13.481 |
| | | 0.11 | **13.523** | 14.063 | **13.618** | 13.783 | **12.770** | 13.293 |
| | | 0.13 | **13.196** | 13.481 | **13.432** | 15.538 | **11.996** | 12.871 |
| | | 0.15 | **12.377** | 13.110 | **12.952** | 15.306 | **11.342** | 12.442 |
| | 5 | 0.00 | 16.464 | **14.786** | **17.705** | 18.601 | **15.227** | 15.675 |
| | | 0.09 | **13.828** | 15.084 | 14.977 | **13.988** | 13.394 | **12.923** |
| | | 0.11 | **13.228** | 13.961 | 14.258 | **12.638** | **12.283** | 12.784 |
| | | 0.13 | **12.701** | 13.631 | 13.498 | **11.540** | **12.144** | 12.360 |
| | | 0.15 | **12.398** | 13.214 | 12.728 | **11.832** | **10.815** | 11.643 |
| | 7 | 0.00 | 16.609 | **14.979** | **14.626** | 16.817 | 15.047 | **12.361** |
| | | 0.09 | 13.962 | **13.819** | 16.286 | **13.567** | 12.818 | **11.953** |
| | | 0.11 | 16.731 | **14.005** | 13.606 | **13.300** | 13.092 | **11.849** |
| | | 0.13 | **12.553** | 13.357 | 18.531 | **12.735** | **11.155** | 11.534 |
| | | 0.15 | **12.232** | 12.996 | 12.726 | **12.316** | **10.577** | 11.768 |
| **ViT** | 12 | 0.00 | **11.577** | 12.124 | **11.577** | 12.124 | **21.308** | 21.793 |
| | | 0.09 | 7.595 | **7.582** | **7.538** | 8.008 | **16.349** | 17.037 |
| | | 0.11 | **7.219** | 7.495 | 7.387 | **7.240** | **16.638** | 18.515 |
| | | 0.13 | **7.094** | 7.138 | **7.021** | 7.351 | **23.186** | 23.482 |
| | | 0.15 | 6.953 | **6.865** | **6.650** | 6.889 | **12.695** | 14.632 |

Table 4: OOD Accuracy AUC values for all language classification settings.

| Model | $L$ | $\gamma_{\text{train}}$ | IMDB | | MNLI | |
|---|---|---|---|---|---|---|
| | | | Constr. | Unconstr. | Constr | Unconstr. |
| | | 0.0 | **1.719** | 1.707 | **0.892** | 0.887 |
| | | 0.2 | **1.719** | 1.714 | **0.902** | 0.896 |
| | 3 | 0.4 | **1.723** | 1.720 | **0.914** | 0.909 |
| | | 0.6 | **1.731** | 1.728 | **0.921** | 0.920 |
| | | 0.8 | **1.740** | 1.735 | **0.930** | 0.926 |
| | | 1.0 | **1.736** | 1.731 | **0.935** | 0.933 |
| | | 0.0 | 1.680 | **1.680** | **0.889** | 0.882 |
| | | 0.2 | 1.686 | **1.687** | **0.902** | 0.894 |
| | 5 | 0.4 | 1.702 | **1.706** | **0.913** | 0.909 |
| | | 0.6 | 1.714 | **1.718** | **0.924** | 0.918 |
| | | 0.8 | 1.727 | **1.729** | **0.930** | 0.927 |
| UT | | 1.0 | 1.739 | **1.740** | **0.933** | 0.931 |
| | | 0.0 | **1.673** | 1.672 | **0.889** | 0.885 |
| | | 0.2 | 1.679 | **1.685** | **0.901** | 0.895 |
| | 7 | 0.4 | **1.696** | 1.695 | **0.915** | 0.911 |
| | | 0.6 | 1.712 | **1.714** | **0.922** | 0.919 |
| | | 0.8 | **1.726** | 1.724 | **0.929** | 0.925 |
| | | 1.0 | **1.736** | 1.733 | **0.932** | 0.931 |
| | | 0.0 | **1.672** | 1.670 | **0.890** | 0.885 |
| | | 0.2 | **1.680** | 1.676 | **0.902** | 0.893 |
| | 9 | 0.4 | **1.697** | 1.693 | **0.914** | 0.909 |
| | | 0.6 | 1.711 | **1.711** | **0.923** | 0.920 |
| | | 0.8 | 1.724 | **1.724** | 0.928 | **0.929** |
| | | 1.0 | **1.734** | 1.733 | **0.932** | 0.930 |
| | | 0.0 | 1.583 | **1.608** | 1.052 | **1.064** |
| | | 0.2 | **1.592** | 1.586 | 1.091 | **1.096** |
| DistilBERT | 12 | 0.4 | **1.653** | 1.602 | **1.162** | 1.153 |
| | | 0.6 | **1.677** | 1.660 | **1.253** | 1.227 |
| | | 0.8 | **1.738** | 1.704 | **1.328** | 1.322 |
| | | 1.0 | **1.789** | 1.745 | **1.408** | 1.405 |
| | | 0.0 | **1.797** | 1.651 | 1.532 | **1.544** |
| | | 0.2 | **1.789** | 1.702 | **1.570** | 1.532 |
| RoBERTa | 24 | 0.4 | **1.785** | 1.642 | **1.621** | 1.568 |
| | | 0.6 | **1.778** | 1.708 | **1.668** | 1.626 |
| | | 0.8 | **1.828** | 1.719 | **1.742** | 1.694 |
| | | 1.0 | **1.874** | 1.824 | **1.783** | 1.756 |

increases, an effect also present on the IMDb dataset, as shown in Section 4. This effect is not uniform across settings, however, as we see that in UT the gap between constrained and unconstrained is largest when training with $\gamma = 0$.

**Infeasible Solutions Leads to Low Performance.** In Figure 11 we present a failure mode where the constrained model failed to converge to a feasible solution, resulting in the constrained model having very low performance across all noise regimes. As mentioned previously, this is is an example that highlights the importance of hyperparameter tuning for the dual problem and verifying that constrained models are feasible at the end of training.

**Effect of training perturbation.** In addition to the OOD analysis on fixed $\gamma_{\text{train}}$, we explore the effect of different training perturbations for DistilBERT on IMDb in Figure 9. On the left plot, we observe that increasing $\gamma_{\text{train}}$ improves robustness at high perturbation levels for constrained models at a slight tradeoff for reduced performance at low test perturbations $\gamma$. For unconstrained models, the tradeoff is not as favorable. On the right plot, we observe smoother descent patterns on the error rate for constrained models.

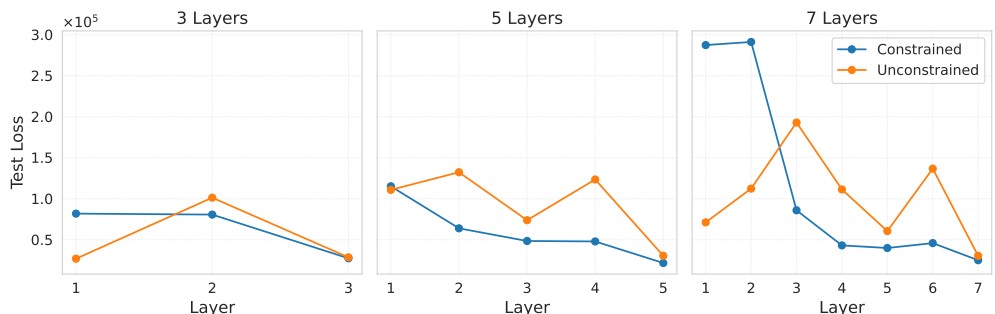

Figure 7: Evolution of the loss function $f$ for unconstrained and constrained DUST models with different numbers of layers: 3, 5, and 7 from left to right. All models are trained on the UCSD dataset with no input perturbations. Constrained DUST exhibits monotonic descent behavior while the unconstrained model does not.

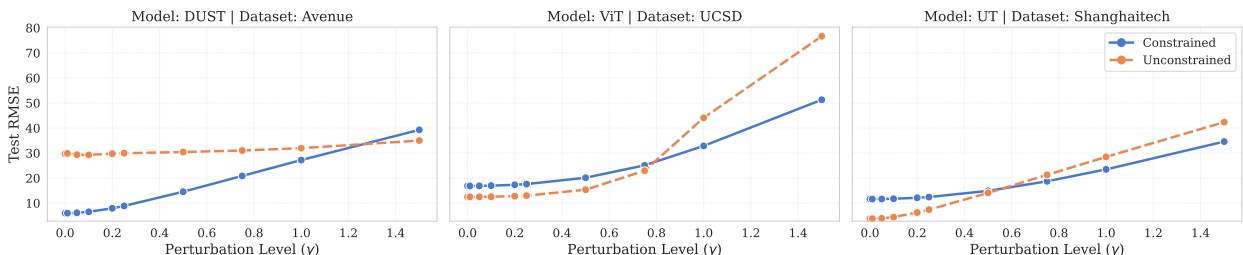

Figure 8: RMSE vs test perturbation plots for constrained and unconstrained models in three select video denoising settings. The first is UT-ShanghaiTech with $L = 7$, train $\gamma = 0.09$, the second is ViT-UCSD with $L = 12$ and training $\gamma = 0.13$, and the third is DUST-Avenue with $L = 7$ and training $\gamma = 0.15$.

### B.5    CONSTRAINT SATISFACTION

**Constraint satisfaction on average.** Figure 13 illustrates the layerwise descent behavior over all training settings. The figures show that constrained models attain consistent, monotonically decreasing patterns.

**Per-sample constraint satisfaction.** Satisfying the descent constraints in expectation of Problem (4) does not guarantee that every sample instance will also satisfy descending constraints. At the time of writing, per-sample constraint satisfaction remains a challenging problem. In light of this, we study the empirical distribution of layerwise ratios. Denoting $f_l^{(m)} = f(\mathbf{X}^{(m)}, \Phi_l(\mathbf{X}^{(m)}))$, we analyze the distribution of $f_l^{(m)}/f_{l-1}^{(m)}$, for all $X^{(m)}$ in a test set of a

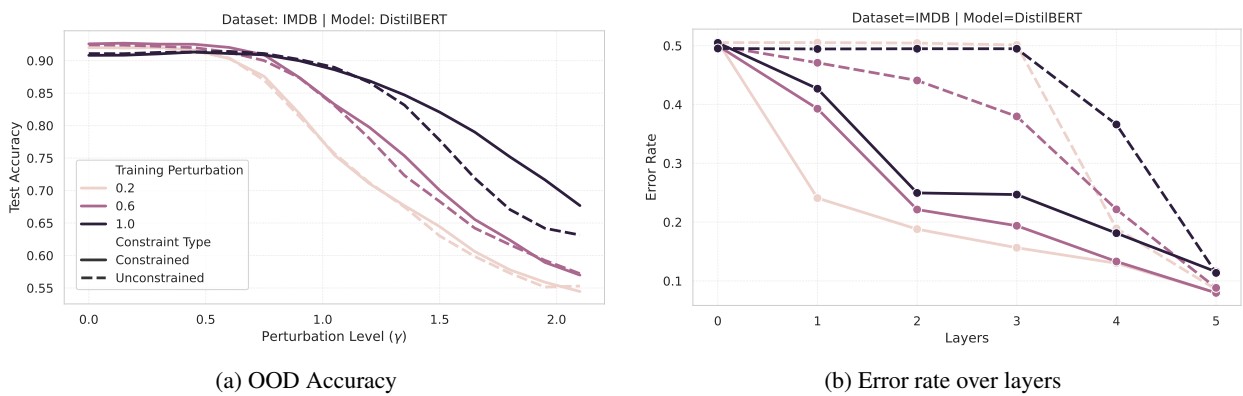

(a) OOD Accuracy                              (b) Error rate over layers

Figure 9: Language Classification OOD Accuracy and layerwise error rates of the constrained and unconstrained DistilBERT trained on the IMDb dataset. Constrained DistilBERT achieves higher OOD accuracy and exhibits monotonic descent behavior across layers for various perturbation levels.

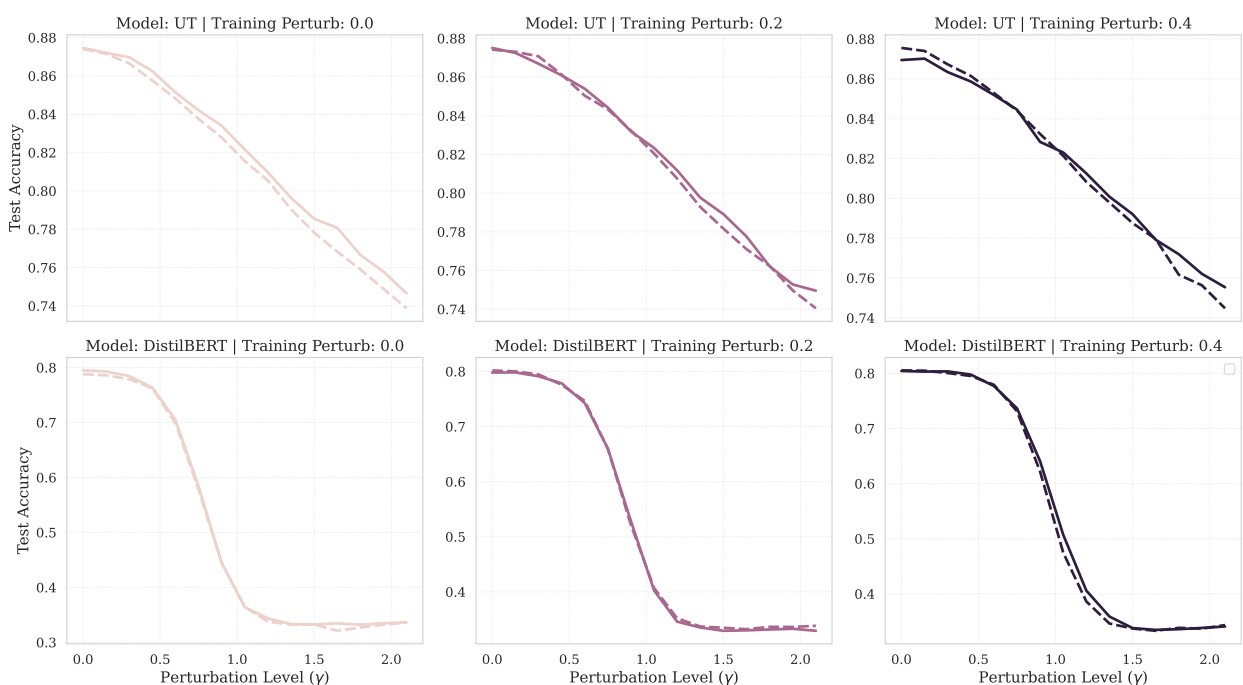

Figure 10: Accuracy OOD plots for two select language experiment settings. The first plot row is UT-IMDB, the second plot row is DistilBERT-MNLI. The plot columns are training $\gamma$ levels. Each plot shows constrained and unconstrained OOD Accuracy curves for each setting. The plot colors represent the training perturbation level.

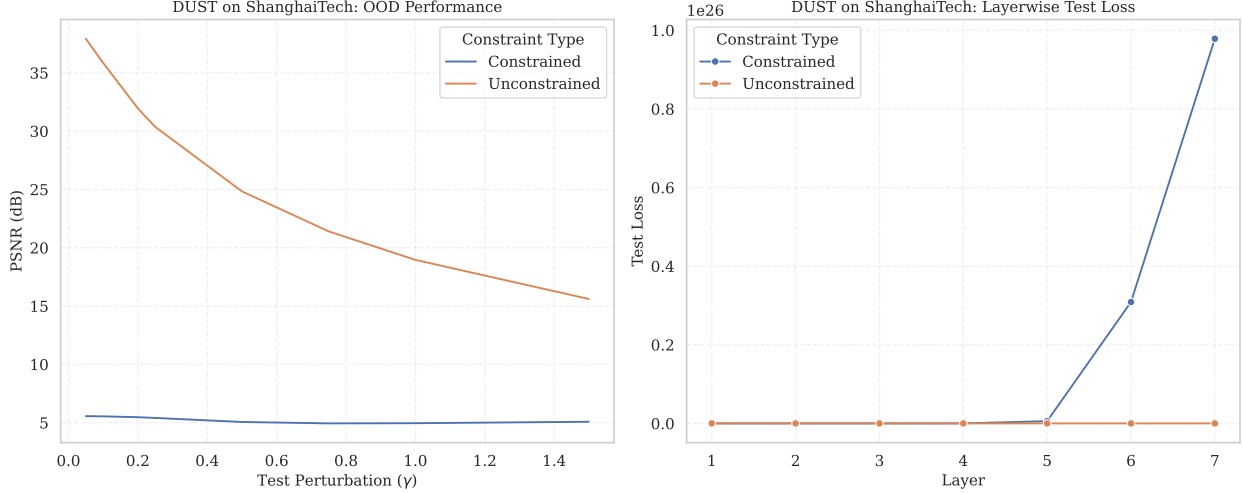

Figure 11: Example of an infeasible constrained model. The setting is DUST on the ShanghaiTech dataset, $L = 7$, training $\gamma = 0.11$. The left plot shows the OOD PSNR values for constrained and unconstrained models, and the right plot shows the test loss of the models at each intermediate layer's representation. The constrained model failed to converge to a monotonically decreasing solution.

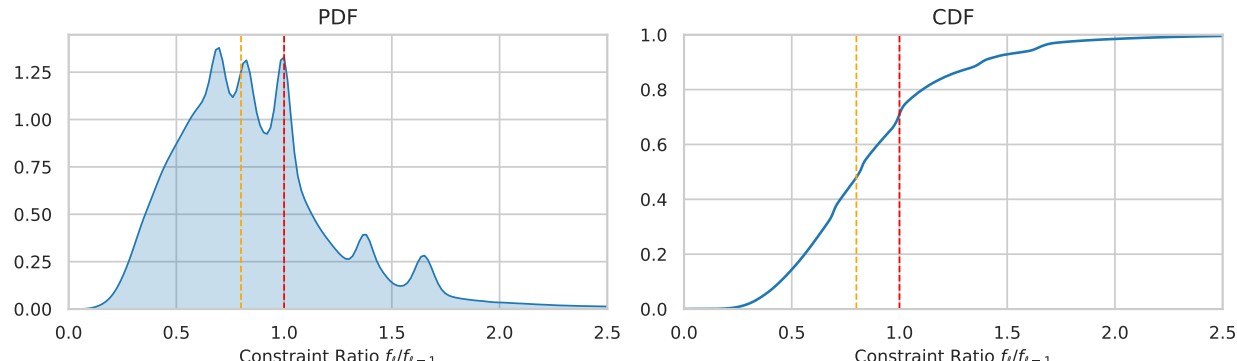

Figure 12: PDF and CDF for empirical layerwise loss ratios. Red line indicates ratio of 1, orange line indicates ratio of $(1 - \alpha) = 0.8$.

constrained model (RoBERTa, IMDb). The constrained model is trained with $\alpha = 0.2$ and without constraint relaxations.

The distribution of layerwise ratios is presented in Figure 12. We observe empirical constraint ratios with mean 0.87 and a median of 0.82. In addition to predictions being concentrated around $(1-\alpha)$, as our theory predicts, we also observe that loss decreases on 70.8% of steps in the constrained case. This shows that empirically, constraints ellicit descent behavior in most steps.

### B.6 ABLATION ANALYSIS SETUP & ADDITIONAL RESULTS

Section 5.1 presented the results of our ablation analysis on the hyperparameters introduced by our constrained learning algorithm. The results were obtained by finetuning RoBERTa on the IMDb dataset, with $\gamma_{train} = 1.0$, for three seeds per ablation setting. For step size, we used values of $\alpha \in \{1.0, 0.98, 0.95, 0.9, 0.8, 0.7, 0.5\}$. For dual learning rate, the values were $\eta_2 \in \{0.001, 0.005, 0.01, 0.05, 0.1\}$. In all cases, set the resilience coefficient $\beta = 1$.

**Ablation on $\alpha$.** Figure 14 shows test accuracy at three different test perturbation levels, with means and error bars calculated across three seeds, for increasing values of $\alpha$. At $\gamma = 0.0$ and 1.05, we observe consistent accuracy levels across all constraint levels (degradation of approximately 0.03%). At $\gamma = 1.65$, test accuracy increases from 65% at $\alpha = 0.0$ to over 75% at $\alpha = 0.5$. This aligns with our theory: increasing the layerwise step size results in improved robustness at a small penalty in in-distribution performance. Finally, we observe increased variance across results at higher perturbation levels.

**Ablation of $\eta_2$.** Figure 15 shows test accuracy at three different test perturbation levels, with means and error bars calculated across three seeds, for increasing values of $\eta_2$. We observe little sensitivity across all values of $\eta_2$. With an appropriately chosen step size to guarantee dual convergence, we can expect similar robustness results.

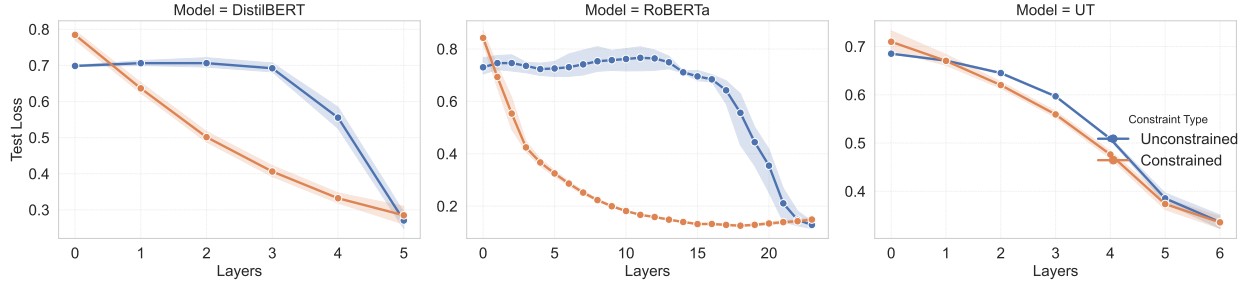

Figure 13: Average layerwise trajectory over all runs of the text classification experiments. UT shows the cases where $L = 7$.

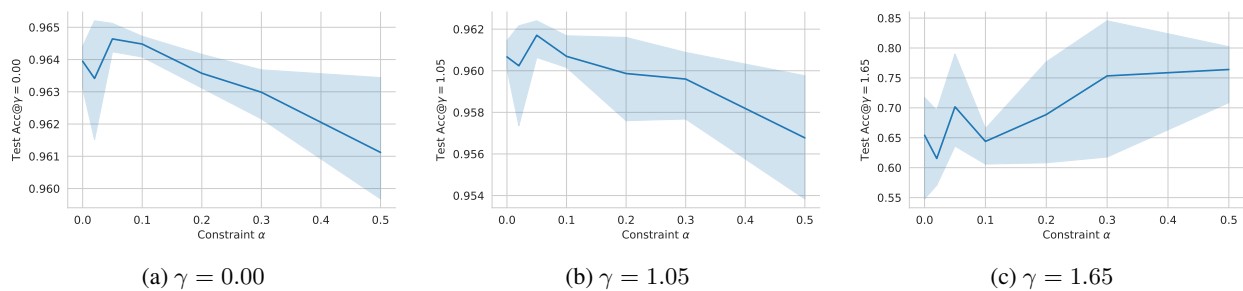

(a) $\gamma = 0.00$  (b) $\gamma = 1.05$  (c) $\gamma = 1.65$

Figure 14: Ablation on constraint $\alpha$. Test accuracy as a function of constraint $\alpha$ for different test $\gamma$. Error bars show variation over three runs.

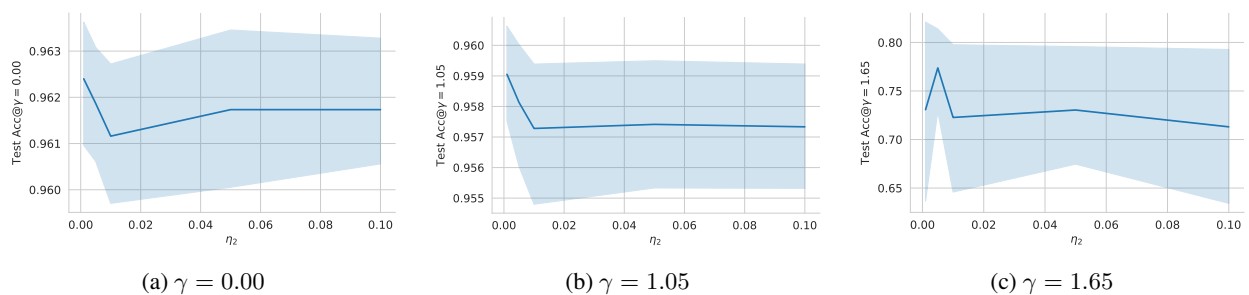

(a) $\gamma = 0.00$  (b) $\gamma = 1.05$  (c) $\gamma = 1.65$

Figure 15: Dual learning rate ablation. Test accuracy as a function of dual learning rate for different test $\gamma$. Error bars show variation over three runs

### B.7 LLM Experiment - Implementation Details & Completions

Both constrained and unconstrained models were fine-tuned with full precision, using AdamW for two epochs, with a batch size of 4. The constrained model used the primal-dual Algorithm 1. OOD robustness was evaluated on a held-out set of the alpaca training data, where the ground truth was perturbed with Gaussian noise. For the Alpaca Eval completions, Gaussian perturbations were added to all tokens, including the model's own generations. Table 5 contains the hyperparameters used for the LLM experiment of Section 6, and 16 presents an example completion from the Alpaca Eval dataset.

Table 5: Hyperparameters for Llama 3.1 8B on instruction-following.

| Model | Dataset | $L$ | $\gamma$ | $\alpha$ | $f_0$ | $\beta$ | $\eta_2$ | $r$ | $\alpha_{\text{LoRA}}$ | dropout |
|---|---|---|---|---|---|---|---|---|---|---|
| Llama 3.1 8B | Alpaca | 32 | 0.25 | 0.95 | 12.0 | 1.5 | $1.00 \times 10^{-1}$ | 6 | 8 | 0.05 |

## C Unrolled Neural Networks

In this Appendix, we discuss the literature on unrolled neural networks, our relationship to unrolling, and implementation details of the unrolled architectures used in the experiments of Sections 4 and 5

### C.1 Algorithmic unrolling and unrolled neural networks

Algorithmic unrolling began with the seminal work by Gregor & Lecun for learning fast approximations of sparse coding (LISTA), which showed that a significant speedup can be accomplished by training a neural network to *imitate* the optimal representation of ISTA. A vast literature has focused on improving the efficiency and convergence of LISTA-like algorithms, for instance, (Liu et al., 2019a).

Since then, a new type of unrolling literature has emerged (Yang et al., 2022; De Weerdt et al., 2023; Yang et al., 2021; Xie et al., 2023; Hershey et al., 2014; Frecon et al., 2022) where neural networks are interpreted as optimizers that settle on an equilibrium point of an energy function. An unrolled network minimizes a problem of the form

**Prompt:** *List the pieces of a reinforcement learning system that can learn how to play Atari games.*

| Unconstrained | Constrained |
|---|---|
| The pieces of a reinforcement learning system that can learn how to play Atari games are: 1. A keyboard, 2. A monitor, 3. A mouse, 4. A monitor 5. A monitor, 6. A monitor, 7. A mouse, 8. A mousse, 9. A mouse, 10. A moose, 11. A moose, 12. A mouse, ... | The reinforcement learning system consists of a set of rules that reward the agent for performing a certain action. These rules are designed to encourage the agent to take certain actions in order to maximize its reward. |

Figure 16: Example completion from Alpaca Eval under $\gamma = 1.5$ embedding noise. The unconstrained model degenerates into nonsensical text, while the constrained model remains coherent.

$$\mathbf{H}^{\star}(\mathbf{W}) = \operatorname*{argmin}_{\mathbf{H}} \mathbb{E}\left[g(\mathbf{X}, \mathbf{H})\right], \tag{39}$$

where $g(\cdot)$ is the energy function, $\mathbf{H}$ is an optimization variable, and $\mathbf{X}$ is data. The network $\Phi(\mathbf{X}; \mathbf{W})$ *unrolls* problem (39) by iteratively decreasing $g$ along its layers. The general method to prove unrolling is to interpret the forward pass as a gradient step, proximal method, or some other descent method,

$$\mathbf{H}_{k+1}(\mathbf{W}) = \mathbf{H}_k(\mathbf{W}) - \eta \Gamma_{g, \mathbf{H}(\mathbf{W})}, \tag{40}$$

where $\Gamma_{\mathbf{H}(\mathbf{D})}$ is a descent direction of $g$ with respect to $\mathbf{H}$. The goal of unrolling is then to find a $g$ function such that $\Gamma_{g, \mathbf{H}(\mathbf{D})}$ makes this equation also satisfy

$$\mathbf{H}_{k+1}(\mathbf{W}) = \phi(\mathbf{H}, \mathbf{X}; \mathbf{W}), \tag{41}$$

where $\phi$ is the forward pass of a neural network, such as a transformer. The motivation is to elucidate the behavior of the neural network, as it is said that the function $g$ *explains* the forward pass of the architecture. For example, consider unrolling the following problem, which results in a ReLU layer:

$$\min_{\mathbf{H}(\mathbf{W})} \frac{1}{2} \mathrm{Tr}[\mathbf{H}^{\top}\mathbf{W}\mathbf{H}] + \frac{1}{2}\|\mathbf{H}\|_2^2 + \psi(\mathbf{H}), \qquad \psi(u) = \begin{cases} +\infty \text{ if } u < 0 \\ 0, \text{ otherwise} \end{cases} \tag{42}$$

unrolls into $\mathbf{H}_{k+1} = \mathrm{ReLU}[\mathbf{W}_s\mathbf{H}_k]$, with the symmetric matrix $\mathbf{W}_s = (1-\alpha)I - \frac{\eta}{2}(\mathbf{W}_2 + \mathbf{W}_2^{\top})$, as shown in (Xie et al., 2023).

**Training unrolled models.** Training an unrolled neural network is given by the following bilevel optimization problem:

$$\mathbf{W}^* = \operatorname*{argmin}_{\mathbf{W}} \quad \mathbb{E}[f(\mathbf{X}, \mathbf{H}^*(\mathbf{W}))], \tag{43}$$

$$\text{s.t.} \qquad \mathbf{H}^*(\mathbf{W}) = \operatorname*{argmin}_{\mathbf{H}} \mathbb{E}[g(\mathbf{X}, \mathbf{H}; \mathbf{W})], \tag{44}$$

where $f$ is a training objective and $g$ is an auxiliary objective function that guides the evolution of the representation $\mathbf{H}$ and encourages desirable internal structure, such as sparsity, cross-correlation, etc.

**Relation to our method.** As we have explained in Section 2, we draw inspiration from the idea of neural network unrolling, but what we call unrolling in this paper is a different method, since we don't design and train neural networks to solve optimization problems, but rather encourage existing architectures to descend on an objective via constraints during training. For standard unrolled transformers, such as DUST and UT, our training method can be seen as solving the bilevel problem (43) and (44). The constraints we impose in Section 3 encourage descent on (43). Since the unrolled model is designed to be a descent algorithm of $g(\cdot)$, its forward pass should descend on 44, by construction.

## C.2 TRANSFORMER UNROLLING

Consider the energy function given by $g(\mathbf{X}, \mathbf{W}) = g_1(\mathbf{X}, \mathbf{W}) + g_2(\mathbf{X}, \mathbf{W})$, with

$$g_1(\mathbf{X}; \mathbf{W}) = -\sum_{t=1}^{T} \sum_{u=1}^{T} \exp\left\{ -\frac{1}{2} \|\mathbf{W}\mathbf{x}_t - \mathbf{W}\mathbf{x}_u\|^2 \right\} + \frac{1}{2} \sum_{t=1}^{T} \|\mathbf{W}\mathbf{x}_t\|_2^2, \tag{45}$$

$$g_2(\mathbf{X}, \mathbf{W}_2) = \frac{1}{2} \mathrm{Tr}\{\mathbf{X}^\top \mathbf{W}_2 \mathbf{X}\} + \frac{1}{2} \|\mathbf{X}\|_{\mathcal{F}}^2 + \varphi(\mathbf{X}) \tag{46}$$

where $\mathbf{W} \in \mathbb{R}^{d \times n}$ is a matrix of learnable parameters. This function consists of a sum of scaled distances between the vectors of the sequence $\mathbf{X}$ and the norm of the projected vectors.

Consider the following recursion that describes a symmetric transformer with shared weights,

$$\mathbf{Z}_{k+1} = \mathbf{X}_k \times \mathrm{sm}\left[ (\mathbf{W}_1 \mathbf{X}_k)^\top (\mathbf{W}_1 \mathbf{X}_k) \right], \tag{47}$$

$$\mathbf{X}_{k+1} = \mathrm{ReLU}\left( \mathbf{W}_s \mathbf{Z}_{k+1} \right), \tag{48}$$

for $k \in [1, K]$, with $\mathbf{W}_s$ a symmetric weight matrix as in (42). Equation (47) is a softmax self-attention layer with a single projection matrix $\mathbf{W}_1^l$ shared between keys, queries, and values, i.e., $\mathbf{Q}^l = \mathbf{K}^l = \mathbf{V}^l$ for all $l$, noting that the value parameters cancel out from the previous layer. This Equation corresponds to the unrolling of (45). In the next section, we will elaborate on the part of the proof from (Yang et al., 2022) that shows how to derive an attention-like structure from this function.

Equation (48) is a linear transformation parameterized by $\mathbf{W}_2$, followed by a residual connection and a ReLU nonlinearity. As mentioned in the previous section, this form of ReLU with symmetric weights corresponds to the unrolling of (46).

With this definition of $g(\cdot)$, (Yang et al., 2022) show that Equations (47) and (48) are a descent algorithm for problem (39). The proof involves showing that both steps sequentially result in an inexact gradient descent direction of $g_1(\cdot) + g_2(\cdot)$.

### C.2.1 DERIVATION OF SOFTMAX SELF-ATTENTION

In (Yang et al., 2022), Theorem 3.1 shows how to derive unrollings for a family of attention structures. Here we present this theorem for the concrete case of self-attention, using our notation. We provide an extended version of their proof for completeness.

**Theorem 1** (Theorem 3.1 from (Yang et al., 2022))**.** *Replace* $\mathbf{Y} = \mathbf{W}\mathbf{X}$*, and consider* $g_1(\mathbf{Y})$ *as in* (45)*. Let* $\boldsymbol{\beta}_i = \exp\left\{ -\frac{1}{2} \left\| \mathbf{y}_i^{(k)} \right\|^2 \right\}$*, and let* $\mathbf{Y}^{(k)}$ *represent any fixed value for* $\mathbf{Y}$*. Then the update rule*

$$\mathbf{y}_i^{(k+1)} = \frac{\sum_{j=1}^{n} \boldsymbol{\beta}_j \exp\left\{ \mathbf{y}_i^{(k)\top} \mathbf{y}_j^{(k)} \right\} \mathbf{y}_j^{(k)}}{\sum_{j=1}^{n} \boldsymbol{\beta}_j \exp\left\{ \mathbf{y}_i^{(k)\top} \mathbf{y}_j^{(k)} \right\}}, \quad \forall i, \tag{49}$$

*satisfies* $g_1\left(\mathbf{Y}^{(k+1)}\right) \leq g_1\left(\mathbf{Y}^{(k)}\right)$ *with equality iff* $\mathbf{Y}^{(k)}$ *is a stationary point of* $g_1$*.*

Replacing $\mathbf{Y} = \mathbf{W}\mathbf{X}$ in (45), we have

$$g_1(\mathbf{Y}) = \sum_{t=1}^{T} \sum_{u=1}^{T} \exp\left\{ -\frac{1}{2} \|\mathbf{y}_t - \mathbf{y}_u\|^2 \right\} + \frac{1}{2} \|\mathbf{Y}\|_{\mathcal{F}}^2. \tag{50}$$

The proof relies on a graph over the tokens, but we will consider the special case of a fully connected graph $G = (\mathcal{V}, \mathcal{E})$,

Let $G = (\mathcal{V}, \mathcal{E})$ a fully connected graph over the tokens, with Laplacian $\mathcal{L} = D - A = B^T B$, where $B \in \mathrm{R}^{m \times n}$ is the incidence matrix Consider the surrogate energy function

$$\tilde{g}_1(\mathbf{Y}, \Gamma) = \sum_{u,v \in \mathcal{E}} \frac{1}{2} \gamma_{u,v} \|\mathbf{y}_u - \mathbf{y}_v\|^2 + \frac{1}{2} \|\mathbf{Y}\|_{\mathcal{F}}^2. \tag{51}$$

Proposition B.2 in (Yang et al., 2022) shows how a majorization-minimization algorithm that decreases $\tilde{g}_1$ also decreases $g_1$. The proof relies on Lemma 3.2 in (Yang et al., 2021). Here we focus on the proof of decreasing $\tilde{g}$.

**Theorem 2** (Theorem B.2 from (Yang et al., 2022)). *Consider updating $\tilde{g}_1$ using a gradient step with step size $\eta$ and Jacobi preconditioner $\mathcal{D}^{-(t)}$:*

$$\mathbf{Y}^{(t+1)} = \mathbf{Y}^{(t)} - \eta \mathcal{D}^{-(t)} \left. \frac{\partial \tilde{g}_1 \left( \mathbf{Y}, \mathbf{\Gamma}^{(t)} \right)}{\partial \mathbf{Y}} \right|_{\mathbf{Y}=\mathbf{Y}^{(t)}}, \tag{52}$$

*where*

$$\mathcal{D}^{(t)} = \left. \frac{\partial^2 \tilde{g}_1 \left( \mathbf{Y}, \mathbf{\Gamma}^{(t)} \right)}{\partial \mathbf{Y}^2} \right|_{\mathbf{Y}=\mathbf{Y}^{(t)}}, \tag{53}$$

*and $\eta \leq 1$, it follows that $\tilde{g}_1 \left( \mathbf{Y}^{(t+1)} \right) \leq \tilde{g}_1 \left( \mathbf{Y}^{(t)} \right)$.*

*And the update rule in (52) can be written as*

$$\mathbf{y}_u^{(t+1)} = (1 - \eta)\mathbf{y}_u^{(t)} + \eta \frac{\sum_{v \in \tilde{\mathcal{N}}(u)} \boldsymbol{\beta}_v \exp \left\{ \mathbf{y}_u^{(t)\top} \mathbf{y}_v^{(t)} \right\} \mathbf{y}_v^{(t)}}{\sum_{v \in \tilde{\mathcal{N}}(u)} \boldsymbol{\beta}_v \exp \left\{ \mathbf{y}_u^{(t)\top} \mathbf{y}_v^{(t)} \right\}}, \quad \forall u. \tag{54}$$

The weights are given by the diagonal matrix $\Gamma$ with entries $\Gamma_{ii} = \gamma_{uv}, \forall\, e_i = (u, v) \in \mathcal{E}$. The $\gamma_{uv}$ correspond to the reweighting coefficients in a minimization-algorithm,

$$\gamma_{u,v}^{(t)} = \frac{d(-\exp\{-\frac{1}{2}\|y_u - y_v\|^2\})}{d(\frac{1}{2}\|y_u - y_v\|^2)} \tag{55}$$

$$= \exp\{-\frac{1}{2}\|y_u^{(t)} - y_v^{(t)}\|^2\} \tag{56}$$

$$= \exp\{y_u^{(t)\top} y_v^{(t)}\}\beta_u\beta_v, \qquad \beta_u = \exp\{-\frac{1}{2}\|y_u^{(t)}\|^2\} \tag{57}$$

Let $\tilde{\mathbf{L}} = B^T \Gamma B$ the reweighted Laplacian (note that $\Gamma \in \mathbb{R}^{m \times m}$ is a diagonal matrix with entries $\gamma_{uv}$ for every arc $(u, v) \in \mathcal{E}$, and that $\gamma_{uu} = e^0 = 1$. The reweighted energy can be written as:

$$\tilde{g}_1(\mathbf{Y}) = \text{Trace}\left[ \mathbf{Y}^T \tilde{\mathbf{L}} \mathbf{Y} \right] + \|\mathbf{Y}\|_{\mathcal{F}}^2 \tag{58}$$

Then its derivative and Hessian are given by

$$\frac{\partial \tilde{g}(\mathbf{Y})}{\partial \mathbf{Y}} = \tilde{\mathbf{L}}\mathbf{Y} + \mathbf{Y} \tag{59}$$

$$\frac{\partial^2 \tilde{g}(\mathbf{Y})}{\partial \mathbf{Y}^2} = \tilde{\mathbf{L}} + \mathbf{I}. \tag{60}$$

Note that the reweighted Laplacian has entries

$$\tilde{\mathbf{L}}_{uv} = \begin{cases} \sum_{v' \neq u} \gamma_{uv'} & \text{if } u = v \\ -\gamma_{uv} & \text{if } u \neq v \end{cases} \tag{61}$$

So the Jacobi preconditioner is the diagonal of the Hessian of $\tilde{g}$, with entries

$$[\mathcal{D}]_{uu} = \sum_{v \neq u} \gamma_{uv} + 1 \tag{62}$$

$$= \sum_{v \in \mathcal{V}} \gamma_{uv} - \gamma_{uu} + 1 \tag{63}$$

$$= \sum_{v \in \mathcal{V}} \gamma_{uv} \tag{64}$$

the equalities coming from adding and subtracting $\gamma_{uu} = e^0 = 1$.

Also note that, considering the two cases of $\tilde{\mathbf{L}}$, we have

$$[\tilde{\mathbf{L}}\mathbf{Y}]_u = \sum_{v' \neq u} \gamma_{uv'}\mathbf{y}_u - \sum_{v \in \mathcal{V}} \gamma_{uv}\mathbf{y}_v \tag{65}$$

The first term coming from the case where $u = v$, and the second term is the sum of all the other cases where $u \neq v$. Replacing this into (52) (and dropping $(t)$ in the right hand side for brevity) we have

$$\mathbf{Y}^{(t+1)} = \mathbf{Y} - \eta\mathcal{D}^{-1}\tilde{\mathbf{L}}\mathbf{Y} - \eta\mathcal{D}^{-1}\mathbf{Y} \tag{66}$$

The update rule for the vector of node $u$, substituting (65) and (64), becomes

$$\mathbf{y}_u^{(t+1)} = \mathbf{y}_u - \eta\frac{\sum_{v' \neq u} \gamma_{uv'}\mathbf{y}_u - \sum_{v \in \mathcal{V}} \gamma_{uv}\mathbf{y}_v}{\sum_{v \in \mathcal{V}} \gamma_{uv}} - \eta\frac{1}{\sum_{v \in \mathcal{V}} \gamma_{uv}}\mathbf{y}_u \tag{67}$$

$$= \mathbf{y}_u - \eta\frac{\sum_{v' \neq u} \gamma_{uv'}}{\sum_{v \in \mathcal{V}} \gamma_{uv}}\mathbf{y}_u - \eta\frac{\gamma_{uu}}{\sum_{v \in \mathcal{V}} \gamma_{uv}}\mathbf{y}_u + \eta\frac{\sum_{v \in \mathcal{V}} \gamma_{uv}\mathbf{y}_v}{\sum_{v \in \mathcal{V}} \gamma_{uv}} \tag{68}$$

$$= \mathbf{y}_u - \eta\left(\frac{\sum_{v \in \mathcal{V}} \gamma_{uv}}{\sum_{v \in \mathcal{V}} \gamma_{uv}}\right)\mathbf{y}_u + \eta\frac{\sum_{v \in \mathcal{V}} \gamma_{uv}\mathbf{y}_v}{\sum_{v \in \mathcal{V}} \gamma_{uv}} \tag{69}$$

$$= (1 - \eta)\mathbf{y}_u + \eta\frac{\sum_{v \in \mathcal{V}} \gamma_{uv}\mathbf{y}_v}{\sum_{v \in \mathcal{V}} \gamma_{uv}} \tag{70}$$

$$= (1 - \eta)\mathbf{y}_u + \eta\frac{\beta_u \sum_{v \in \mathcal{V}} \exp\{-\mathbf{y}_u^T\mathbf{y}_v\}\beta_v\mathbf{y}_v}{\beta_u \sum_{v \in \mathcal{V}} \exp\{\mathbf{y}_u^T\mathbf{y}_v\}\beta_v} \tag{71}$$

$$= (1 - \eta)\mathbf{y}_u + \eta\frac{\sum_{v \in \mathcal{V}} \exp\{\mathbf{y}_u^T\mathbf{y}_v\}\beta_v\mathbf{y}_v}{\sum_{v \in \mathcal{V}} \exp\{\mathbf{y}_u^T\mathbf{y}_v\}\beta_v} \tag{72}$$

which is (54). This completes the proof of Theorem 2. Note that convergence requires that $\alpha \leq 1/L^{(t)}$, with $L^{(t)}$ the Lipschitz constant of the gradient of $\mathcal{D}^{-(t)}\tilde{(}\mathbf{Y}, \mathbf{\Gamma}^{(t)})$, but $L^{(t)} = 1$.

**Proof of Theorem 1.** With $\eta = 1$ the $y_u$ term vanishes, replacing the summation over all vertices with the vertex indices, and setting the temperature parameters to $\beta_u = 1$ for all $u \in \mathcal{V}$, Equation 72 becomes (49). By Theorem 2, this update rule decreases $\tilde{g}(\cdot)$.

Furthermore, Proposition B.2 in (Yang et al., 2022) shows how a majorization-minimization algorithm that decreases $\tilde{g}_1$ also decreases $g_1$. The proof relies on Lemma 3.2 in (Yang et al., 2021).

**SSA Reparametrization.** If we replace back $\mathbf{y}_u = \mathbf{W}\mathbf{x}_u$ in (49), we have, in matrix form,

$$\mathbf{W}\mathbf{X}^{(k+1)} = \mathbf{W}\mathbf{X}^{(k)} \cdot \text{sm}\left[\left(\mathbf{W}\mathbf{X}^{(k)}\right)^\top \left(\mathbf{W}\mathbf{X}^{(k)}\right)\right], \tag{73}$$

where we see that $\mathbf{W}$ cancels out on both sides, leading to what we call symmetric softmax attention without a $\mathbf{V}$ matrix.

## C.3 Deep Unfolded Sequential Transformer (DUST)

Leveraging the transformer unrolling seen in the previous section, Deep Unfolded Sequential Transformer (DUST) (De Weerdt et al., 2023) derives an architecture for video processing by unrolling the function

$$\min_{\mathbf{H}} \quad \mathbb{E}\Big[\underbrace{\sum_{t=1}^{T}\Big(\tfrac{1}{2}\|\mathbf{x}_t - \mathbf{ADh}_t\|_2^2 \;+\; \lambda_1\|\mathbf{h}_t\|_1\Big)}_{\text{LISTA loss}}\Big] \;+\; \lambda_2\, g_1(\mathbf{H}, \mathbf{D}), \tag{74}$$

where $g_1$ is the symmetric attention energy as in Equation (45), $\mathbf{X}$ is a sequence of video frames as defined in Section 4, $\mathbf{D} \in \mathbb{R}^{D \times N}$ is the feature dictionary, $\mathbf{A} \in \mathbb{R}^{m \times D}$, $m \ll D$ is the measurement matrix, $\mathbf{H} \in \mathbb{R}^{N \times T}$ is called the *sparse code*, and $\lambda_1$ and $\lambda_2$ are regularization parameters. Unrolling this problem results in a symmetric softmax attention layer followed by a LISTA layer. The $g$ function is composed of three components: the objective $f$, $\ell_1$ terms to promote sparsity, and the cross-correlation term $g_1$. With this structure, DUST aims to learn a sparse reconstruction $\mathbf{H}$ of the dictionary features $\mathbf{D}$ that also takes into account the cross-correlation of elements in the sequence.

The DUST architecture is

$$\mathbf{H}^{(k+1/2)} = \lambda_2 \mathbf{H}^{(k)} \operatorname{softmax}(\mathbf{H}^{(k)\top} \mathbf{D}^\top \mathbf{D} \mathbf{H}^{(k)}), \tag{75}$$

$$\mathbf{H}_t^{(k+1)} = \phi_{\frac{\lambda_1}{c}}\left(\mathbf{U}\mathbf{H}_t^{(k+1/2)} + \mathbf{VX}\right), \tag{76}$$

for all $k \in [1, K]$. Here, the proximal operator $\phi_\gamma(u) = \operatorname{sign}(u)\max(0, |u| - \gamma)$ is called the soft-thresholding function, and $c$ is related to the Lipschitz constant of the gradient of $\mathbf{X}-$ The index $(k + 1/2)$ denotes an intermediate step.

In (De Weerdt et al., 2023), $\mathbf{U}, \mathbf{V}, \mathbf{D}, c, \lambda_1$ and $\lambda_2$ are learnable parameters. The matrices $\mathbf{U}$ and $\mathbf{V}$ are initialized as $\mathbf{U} = \mathbf{I} - \frac{1}{c}\mathbf{D}^\top \mathbf{A}^\top \mathbf{A}\mathbf{D}$, and $\mathbf{V} = \frac{1}{c}\mathbf{D}^\top \mathbf{A}^\top$, while the dictionary $\mathbf{D}$ is initialized to the Discrete Cosine Transform (DCT) and updated during training as well.

Alternate execution of the updates (75) and (76) is guaranteed to reduce the objective of (74) leveraging the same Alternate Minimization results from (Yang et al., 2022).

## C.4 Implementation Adaptations from DUST

We make the following departures from the original paper's implementation:

**No compressed sensing.** The experiments of (De Weerdt et al., 2023) include compressed sensing tasks. In these experiments, the measurement matrix $\mathbf{A}$ that compresses the signal into a lower dimensional space. This matrix is included in the unrolling of the LISTA layer to account for this signal compression. Since we only work with the denoising task, we set $\mathbf{A} = I$ in our architecture.

**Coupled LISTA weights.** We keep the coupling of $\mathbf{U}$ and $\mathbf{V}$ within each layer instead of learning them, motivated by results from the LISTA literature that suggest the coupling of these matrices is necessary for convergence (Chen et al., 2018).

**Disabling some learnable parameters.** In our implementation, the only trainable parameter is $\mathbf{D}$ for unconstrained DUST and $\mathbf{D}^l$ for all $l$ in constrained DUST. We fix the learnable parameters to $\lambda_1 = 0.9$, $\lambda_2 = 0.25$, as we observed the learnable parameters resulted in trivial satisfaction of our descent constraints (by making one term have zero weight). Preliminary experiments suggested this came with little impact to performance for unconstrained runs.

**Decoupled Layerwise Dictionaries.** Although (Liu et al., 2019a) advocates parameter coupling across layers, we maintain distinct dictionaries $\mathbf{D}$ for each layer in constrained models, as we empirically observed this improves performance.

