# OpenReview forum: "A Constrained Optimization Perspective of Unrolled Transformers"
_ICLR.cc/2026/Conference — Submitted to ICLR 2026_

### Official Review · Reviewer_96jF · 2025-10-22

**Soundness:** 4
**Presentation:** 4
**Contribution:** 4
**Rating:** 8
**Confidence:** 3

**Summary:**

The authors consider the theory and practice of training deep neural networks to iteratively improve the value of a self-supervised loss function _at each layer_ (as opposed to end-to-end training which only considers the network's output at the last layer). They prescribe that the improvement should be at a geometric rate, which gives a constrained optimization problem for the network parameters. They propose and justify the use of primal-dual methods for optimizing this constrained problem by bounding the duality gap under some assumptions (Theorem 1). They further bound the performance gap between the network constructed by solving this constrained problem and the optimal network, under the same set of assumptions, on both an in-domain distribution from which the training set is drawn (Theorem 2) and an out-of-domain distribution (Theorem 4). They do some experiments using video denoising and text classification to attempt to demonstrate that the method is reasonable in practice.

**Strengths:**

- The setting and problem formulation is novel; as far as I know, explicitly end-to-end training neural networks for layer-wise behavior has not been done before. However,
- The method is clean, well-motivated, and backed directly by the theory.
- The empirical results look promising: while the performance on the original task does not degrade (as one might expect by making the model make trade-offs by adding more terms to the objective), each layer indeed does optimize the objective incrementally, adding some basic understanding of the network and robustness/generalization capabilities.
- The theory is also good; the assumptions made are mostly realistic, i.e., the loss is assumed to be a convex function of the data and prediction, not of the network parameters, yet optimization results can still hold for the network parameters. The analysis is also clean and uses some clever techniques (i.e., working with supermartingales).
- Lastly, the paper is well-written and easy to read.

**Weaknesses:**

The only issue I have is that the experiments do not quite have the breadth that the generality of the framework demands. Namely, the proposed learning method seems like it would apply in general to all types of self-supervised learning, including diffusion, LLMs, etc., and the text embedding experiment demonstrates a way that it can work for supervised learning too. It would be super interesting to see how the method works on these very popular tasks; if the answer is "it works well" that would strengthen the paper's main claims, but if the answer is "it does not work well" that would still be very interesting and a deeper understanding of how it would break down would also strengthen the paper.

**Questions:**

Q1: See above "Weaknesses" section.

Q2: Does the theory hold straightforwardly for the case where there is side information such as labels? The text embedding experiment seems to suggest this, but I wonder if it is easy to extend the theory. Even better if it accommodates that the loss is arbitrarily dependent on the label (since it's not being fed through the network).

Q3: Have you tried probing the interpretability of these specially trained models? Say via auto-interpretability methods or just by hand. I think it would be really interesting to see how the features evolve differently in conventionally trained networks versus these specialized ones.

---

> ### Author Response · Authors · 2025-11-25
>
> We are very grateful to the reviewer for their positive comments regarding our work. As correctly pointed out, our method is general enough to be applied to self-supervised learning, Diffusion, LLM finetuning, etc. Each of these applications has particularities that make them exciting future directions for this work, as they would require careful experimental validation. To partially address this limitation, we have conducted an additional experiment of LLM supervised fine-tuning on an instruction-following dataset. There, we observe a similar robustness pattern for a Llama-8B model, confirming that the method can be translated to other tasks and larger scales.
>
> Q2: Yes, the reviewer's intuition is correct.  A transformer attached to a classifier head, trained with a cross-entropy loss (such as in the text embedding experiment, see line 363), is covered by our theory, since it satisfies our assumptions on $f$ (see A.1, $f$ is $C$-Lipschitz, continuous, and bounded). This is true in both the text classification case (Section 5) as well as the next-token prediction case. We also provide a new LLM finetuning experiment, where constrained models show improved robustness with marginal ID degradation, as measured by token accuracy and preference in Alpaca Eval with a GPT-5 judge Please see Section 6, Figure 6 and Appendix B.7 in the updated manuscript for more details.
>
> Q3: This is indeed a valuable future direction of work. It would be interesting to further explore whether there exists a qualitative difference in the features learned by constrained models. Regardless, imposing descent constraints also enables a degree of interpretability of the model's behavior: we are training a transformer to behave as an optimization algorithm of the objective.

---

> > ### Comment · Reviewer_96jF · 2025-11-26
> > **Reply to Authors**
> >
> > Thanks for the reply and the additional experiments on LLMs which also serves to demonstrate the potential of the method at larger scales (even if it is "just" for fine-tuning). This makes the work more solid and relevant to practitioners.
> >
> > Re: the answer to Q2: Training a classifier would be interesting for sure, but I was also wondering about the case where the loss is dependent on the label in a more complex way e.g., conditional video denoising given a text prompt. If the framework extends to this setting, it should apply to many reasonable and practical deep learning tasks. This setting would probably require more extensive mathematical analysis and could be postponed to future work, but do you think it is reasonable that such results could exist?
> >
> > I will keep an accept score to reflect the strength of the work.

---

### Official Review · Reviewer_5CtJ · 2025-10-31

**Soundness:** 3
**Presentation:** 2
**Contribution:** 2
**Rating:** 4
**Confidence:** 3

**Summary:**

This paper introduces a framework for training transformers by imposing layerwise descent constraints on the expected loss across layers. Instead of standard empirical risk minimization, the authors formulate transformer training as a constrained optimization problem solved via a primal-dual approach. The key idea is to make each transformer layer behave like a step in a descent algorithm, thereby ensuring monotonic loss reduction through the depth of the network. The authors adopt theory and the objective from [1] to design such constrained Transformer training, and demonstrate empirical improvements on video denoising and text classification (IMDb, MNLI) tasks.

**Strengths:**

1. **Novely:** Integrating constrained training objective to transformers is an interesting touch with motivation from the success of traditional unrolled neural models.
2. **Theoretical foundation:** Although not a brand new contribution, the framework is based on a fairly well-established constrained learning framework of [1] and apply it to the Transformer architecture.
3. **Empirical support:** The effectiveness of the framework is supported with positive empirical results in video denoising and text classification together with OOD tests.

**Weaknesses:**

1. **Scope:** the constrained learning framework seems to be architecture agnostic in most ways. This would mean most theoretical as well as empirical results should ideally be true across any deep neural networks. I think this needs to be discussed in the main text.
2. **Applicability:** would OOD gains achieved with the method scale with model size? Since experiments are mostly small scale, this is not evident if just scaling the model size would overshadow the OOD benefits of the method.
3. **Ablation study:** the paper lacks ablation study on hyperparameters such as $\alpha$ and dual learning rate $\eta$.

**Questions:**

1. **Applicability:** As true for most new pretraining methods, it might be hard to train and test on a very big scale. However, with this method, I wonder if fine-tuning a pre-trained model could benefit too. Could authors validate if fine-tuning on a new task using constrained learning objective could, for example, enjoy boost in OOD performance or generalization? I believe this would make the method more appealing to wider group of readers.
2. How sensitive is the performance to the choice of abovementioned hyperparameters?

___

Overall, I think the paper is interesting and adds a valuable contribution to the ICLR community and, therefore, I am open to increase my score as long as both of my questions are addressed in a satisfactory manner.

___

### References
1. Luiz F. O. Chamon, Santiago Paternain, Miguel Calvo-Fullana, and Alejandro Ribeiro. Constrained Learning with Non-Convex Losses. IEEE Transactions on Information Theory, 2023

---

> ### Author Response · Authors · 2025-11-25
>
> **Response to weaknesses**
>
> 1. **Scope**: We appreciate the reviewer's observation. It is indeed true that our results hold in general for other architectures as well. We have added an observation at the end of Section 2 to account for this fact.
> 2. **Applicability:** This is a good question. Our experiments suggest that yes, OOD gains are still present as model size scales. In addition to the robustness gains in DistilBERT (66M) and RoBERTa (355M), we have added a supervised finetuning experiment with LLMs using Llama 8B and the Alpaca dataset, where constrained models also showed improved robustness with marginal ID degradation, as measured by token accuracy and preference in Alpaca Eval with a GPT-5 judge. Please see Section 6 and Appendix B.7 for details.
> 3. **Ablation study:** Done. We have updated the paper with an ablation analysis on the dual learning rate and constraint parameter $\alpha$, for a fixed setting in the text classification experiment. Please see Section 5.1, Figure 5, and Appendix B.6 for more details. The main takeaways from this analysis:
>    1. OOD increases with a higher constraint parameter $\alpha$.
>    2. ID *marginally* decreases (less than 0.1% decrease in accuracy) with a higher $\alpha$.
>    3. Ablations on the dual learning rate $\eta_2$ show little sensitivity in terms of ID/OOD performance.
>
> **Response to the questions**
>
> 1. Indeed, addressing this question for pretraining is beyond our computational budget. However, to partially respond to the question, we refer to our new LLM experiment, ID performance is maintained while boosting OOD robustness. There is reason to believe that pretrained models would similarly benefit from training with descent constraints.
> 2. We find little sensitivity to the dual learning rate in terms of ID/OOD performance. For the constraint $\alpha$, we see a behavior consistent with our theory:  larger $\alpha$, leads to an increase in OOD performance, and a marginal reduction of ID performance.

---

> > ### Comment · Reviewer_5CtJ · 2025-11-26
> > **Concerns mostly addressed**
> >
> > Thanks for the additional discussion and experiments. I am raising my score accordingly.

---

### Official Review · Reviewer_nzeY · 2025-11-01

**Soundness:** 3
**Presentation:** 3
**Contribution:** 3
**Rating:** 4
**Confidence:** 3

**Summary:**

This paper proposes a constrained optimization framework for training transformers that behave like iterative descent algorithms. Instead of relying only on empirical risk minimization (ERM), the authors introduce layerwise descent constraints, ensuring that each layer of the transformer monotonically decreases the expected loss. Training is performed using a primal-dual algorithm, which enforces these constraints and yields transformers with improved robustness and generalization.

**Strengths:**

- The paper introduces a constrained optimization view of transformer training, in which each layer must monotonically reduce the expected loss—a property inspired by iterative optimization algorithms.
- It formalizes this idea rigorously using a primal–dual training framework, backed by proven results such as:
  - Convergence guarantees (Theorem 2)
  - Out-of-Distribution (OOD) generalization bounds (Theorem 4)
  - The inclusion of expressivity and sample complexity terms (ν, ζ(M, δ)) provides an interpretable bridge between deep-learning practice and optimization theory.

- The authors test their framework on two modalities:
  - Video denoising
  - Text classification with perturbed embeddings
  - Across both domains, the proposed method shows better performance on robustness and generalization.

**Weaknesses:**

**1. Sacrificing in-domain performance**

Figure 2 indicates that the proposed constrained‑optimization transformer underperforms compared to the vanilla ERM baseline on in‑domain (ID) evaluation, while providing advantages mainly in out‑of‑domain (OOD) settings.

This gap suggests that the imposed per‑layer descent constraints may introduce an inductive bias that prioritizes generalization robustness at the expense of ID accuracy. While this trade‑off can be acceptable in robustness‑critical regimes, it raises questions about the method’s scalability in web‑scale scenarios — where data distributions are broad and effectively in‑domain, and OOD events are relatively rare.

Standard regularization / early stopping methods also sacrifice ID performance for OOD generalization. These simple baselines are not compared in this paper.

**2. No causal or autoregressive extension**

The paper develops its constrained optimization formulation exclusively for bidirectional or sequence‑to‑sequence transformer settings, where the entire input sequence is jointly processed at each layer. However, the formulation does not address causal masking or autoregressive factorization of likelihoods, which are crucial for generative models such as GPT‑style architectures.

**3. Scaling**

The paper does not investigate how the proposed constrained‑optimization approach behaves when model size, dataset size, or sequence length scale up. Key scaling questions remain open, such as:

- How the primal–dual updates scale with model parameter / dataset size?
- Whether the constraint coefficients or dual regularizers need retuning for larger models?
- How total compute and memory overhead of per‑layer constraint evaluation grow in long‑sequence or high‑parameter settings?

The current experiments use relatively small encoders (e.g., DistilBERT) and medium‑sized video transformers, leaving unclear whether the method remains efficient or stable for foundation‑scale models.

**4. Writeup tiny issue**

There are many metrics in this paper and some are higher-is-better while others are the opposite. It would be better if it's indicated by $\uparrow$ $\downarrow$ in each figure.

**Questions:**

- Does the proposed method

---

> ### Author Response · Authors · 2025-11-25
>
> 1. **ID performance.** The reviewer raises a valid point: there are many cases where ID degradation may not be acceptable. What we observe in experiments is that for larger models (DistilBERT, RoBERTa, now also Llama 8B), ID seems to degrade less, and OOD improvements are larger. For instance, in our new Llama 8B experiment, constrained Llama has a length-controlled win rate of 50% versus unconstrained in-distribution, and a 69.7% win rate in the presence of input noise (see Section 6, and Figure 6 in the main body, and Appendix B.7). This result suggests that constraints preserved the ID performance of the LLM, while improving robustness. It corroborates our intuition of model capacity: a model with higher capacity has the potential to improve both ID performance and OOD performance, with ID performance improving in both training paradigms, and OOD improving only if the model is specifically trained to do so (i.e., with constraints). Smaller models appear to be more prone to making tradeoffs.
> 2. The reviewer is correct to point out that we do not directly address the autoregressive case. Our theory still holds for next-token predictors trained on cross-entropy, thanks to the weak assumptions made on $f$ (see A.1, $f$ is $C$-Lipschitz, continuous, bounded). This is also supported by our new experiment on Llama 8B.
> 3. We address each question separately:
>    - **Dataset size:** The duality gap of Theorem 1, and the convergence bounds of Thm 2, Cor 3, and Thm 4, depend on sample complexity $\zeta(M,\delta)$, which can be reduced with a larger sample size $M$. This means that the error should decrease as the dataset size grows.
>    - **Model parameters:** Relative to regular training, we only require an additional $L$ parameters for the dual variable vector $\boldsymbol{\lambda}$.
>    - **Compute and memory overhead:** Relative to regular training, the additional computational and memory costs are small. Memory-wise, $\lambda$ is a vector of dimension $L$. Computationally, the cost scales with $L$. To see this, note that the only new operations (relative to standard training) are: computing the penalties for the Lagrangian in (5) and the dual gradient step in Algorithm 1 Line 8\. Both of these operations reuse the forward pass activations $\Phi_l(\mathbf{X};\mathbf{T})$. Given that even the largest LLMs rarely have more than 300-500 layers (e.g. Llama 3.1 405B has 126 layers), this cost is negligible relative to conventional training. For reference, one epoch of our unconstrained Llama finetuning takes 3h51m, while constrained took 3h56m (+2% increase in total training time). Finally, the additional inference cost is zero, because constraint evaluation is not necessary at this stage.
>    - **Tuning:** In principle, yes, the constraint step size $\alpha_l$  needs retuning for models with a larger number of layers. However, as the ablation analysis shows, constrained models are not overly sensitive to the choice of step size, thanks to resilient relaxations. See the updated PDF, Section 5.1 and Appendix B.6.
>    - **Efficacy of our method for larger models.** This is a valuable suggestion that was shared among other reviewers. For this reason, we have prepared an additional experiment for supervised finetuning of a Llama 8B model on an instruction following dataset (Alpaca) and performed a similar OOD analysis. We find that our method scales and provides benefits to larger models with no perceptible ID degradation, as measured by test token accuracy and the preferences of a GPT-5 judge on Alpaca Eval. See below for more details.
> 2. **Writeup:** Done. Thank you for pointing this out. Indeed, not all plot captions had the arrow indicators.

---

### Official Review · Reviewer_sCkt · 2025-11-03

**Soundness:** 3
**Presentation:** 3
**Contribution:** 2
**Rating:** 4
**Confidence:** 3

**Summary:**

This paper proposes a constrained optimization framework for training transformers, such that each layer optimizes an optimization objective and decreases the associated loss function in expectation. From this perspective, the so-trained transformers are also called constrained “unrolled” transformers, as each layer can be interpreted as an unrolling step of the underlying optimization problem. To train such models, this paper introduces a dual training algorithm and provides theoretical insights on the effectiveness of the proposed algorithm, particularly on robustness under distribution shifts. Empirically, the authors evaluated and demonstrated the effect of adding such constraints on tasks including video denoising and text classification.

**Strengths:**

1. This work focuses on decreasing the expected loss value along the layers of the transformer models. This is novel compared to prior work, where most implement each layer as a gradient descent step of the optimization objective. This makes the proposed method work across different transformer architectures.
2. A rather principled and theoretically motivated approach that provides certain performance guarantees. As a bonus, the algorithm itself is also simple enough and is generally applicable.
3. The paper is mostly well-written and clearly presented.

**Weaknesses:**

1. While the authors rightfully state that “the behavior of these networks [from previous works] is non-monotonic along the iterates”, the proposed constrained optimization algorithm only applies to the expectation level rather than sample level. Hence, there is no guarantee that the network from the proposed algorithm will behave monotonically in a real-world setting of finite, streaming samples.
2. Another weakness concerns the experimental results. In the video denoising setting, only 5 out of 9 settings (also noted by the authors) show an advantage for the proposed algorithm. This raises questions about the real-world potential of the proposed algorithm.
3. The experiments focus on distribution shifts, but the type of shifts are rather limited, with Gaussian embedding/pixel noise. To really showcase the strength, other types of shifts (e.g. temporal, non-Gaussian) should also be considered.
4. Another important missing piece is the layer-wise loss analysis. Since the algorithm implements a layer-wise optimization procedure, it is important to experimentally corroborate whether the real practice matches with the theories.

**Questions:**

1. Could you clarify what the U matrices in equation 2 represent? This layer should just be the MLP if I understand correctly.
2. How sensitive is the $\lambda$ parameter and how much tuning is required?

---

> ### Author Response · Authors · 2025-11-25
>
> **Response to weaknesses**
>
> 1. Correct, we train and provide guarantees in expectation. Indeed, guaranteeing feasibility per sample is a much more challenging problem. In practice, however, we observe that, in addition to the constraints being satisfied in the average, the loss decreases in 70.8% of *all steps* across *all samples* of the test set, indicating that for most individual samples the constraints still elicit descent behavior.  We have included a more detailed exploration of the per-sample constraint satisfaction distribution in Appendix B.5. It is also important to emphasize that what matters is making progress across the layers for each example, not enforcing descent at every single layer. Stochastic gradient descent is a classical example, where “mistakes” occasionally happen along its trajectory, but it still converges because these errors do not accumulate and it continues to make overall progress toward the optimum. The performance gains we observe in our numerical results suggest that an analogous phenomenon occurs under our expectation-based constraints.
> 2. The reviewer is correct, we improve in five out of nine cases in video. We interpret Figure 2 as: improving in 5 cases, perform similarly in two, and slightly degrade in two ( 5W 2L 2T), which means we remain comparable or improve robustness in most cases but the benefits of descent constraints may still be task/dataset/architecture dependent. In addition to this, the applicability is highlighted by the consistently positive results we observed in language experiments.
> 3. The reviewer makes a fair point. Considering other types of noise is a valuable future direction (temporal shifts as the reviewer points out, Laplacian perturbations, diffeomorphisms, etc). It is worth pointing out, however, that robustness to Gaussian perturbations supports a wide range of applications (e.g. Differential Privacy, Adversarial Robustness, etc.).
> 4. Also a good point. We agree with the reviewer that layerwise loss is central to our paper. For this reason, we have added a more focused presentation of layerwise loss patterns across run settings in Appendix B.6. Notice that we also provide specific instances of the layerwise loss patterns of constrained and unconstrained models in Figure 1, the left-hand layerwise plot of Figure 5, and Appendix B.4 - “Monotonic Descent Behavior”, Figures 7, 8, 9.
>
> **Response to questions**
>
> 1. This can be interpreted as having a parameterized residual connection. Indeed, if we set  $U=I$, then Equation (2) becomes an MLP with a residual connection, which is how most transformers in the paper are implemented (DistilBERT, RoBERTa, UT). The reason for using the formulation in (2) is to provide a more general framework that accounts for the DUST architecture (see Appendix C.3, Equation 76), which has two learnable matrices due to a LISTA nonlinearity.
> 2. We have added an ablation analysis of the dual hyperparameters that govern the dual variable $\lambda$ in our updated manuscript (Section 5.1 and Appendix B.7). We find low sensitivity to the dual learning rate $\eta_2$ and a predictable effect on $\alpha$.

---

### Meta-Review · Area_Chair_za6d · 2026-01-07

**Summary:**

This paper proposes a constrained optimization framework for training transformers, which enforces layerwise descent constraints on the objective function.  The benefit of the proposed method is stronger robustness to perturbations and maintain higher out-of-distribution generalization. Here are some concerns raised by reviewers.

1. Experimental results do not show an advantage for the proposed algorithm.
2. The experiments on distribution shifts only focus on limited type of shifts.
3. Lack layer-wise loss analysis
4. Sacrificing in-domain performance
5. Focus exclusively on bidirectional or sequence‑to‑sequence transformer settings; no causal or autoregressive extension.
Lack scaling analysis: experiments are mostly on small models.

**Reviewer Concerns:**

The concerns on the sufficiency of experimental studies are not fully addressed. The paper makes some interesting points, but the current empirical results are not sufficiently convincing.

**Reviewer Scores:**

4, 4, 4, 8

---

### Decision · Program_Chairs · 2026-01-26

Reject